# Set4 regulates stress response genes and coordinates histone deacetylases within yeast subtelomeres

Yogita Jethmalani[1], Khoa Tran[1], Maraki Y Negesse[1] , Winny Sun[1], Mark Ramos[2] , Deepika Jaiswal[1], Meagan Jezek[1] , Shandon Amos[1], Eric Joshua Garcia[1], DoHwan Park[2] , Erin M Green[1] 

**The yeast chromatin protein Set4 is a member of the Set3-subfamily of SET domain proteins which play critical roles in the regulation of gene expression in diverse developmental and environmental contexts. We previously reported that Set4 promotes survival during oxidative stress and regulates expression of stress response genes via stress-dependent chromatin localization. In this study, global gene expression analysis and investigation of histone modification status identified a role for Set4 in maintaining gene repressive mechanisms within yeast subtelomeres under both normal and stress conditions. We show that Set4 works in a partially overlapping pathway to the SIR complex and the histone deacetylase Rpd3 to maintain proper levels of histone acetylation and expression of stress response genes encoded in subtelomeres. This role for Set4 is particularly critical for cells under hypoxic conditions, where the loss of Set4 decreases cell fitness and cell wall integrity. These findings uncover a new regulator of subtelomeric chromatin that is key to stress defense pathways and demonstrate a function for Set4 in regulating repressive, heterochromatin-like environments.**

## Introduction

The regulation of gene expression in response to changing environmental signals is dependent on a diverse set of chromatin-binding proteins, including transcription factors, histone-modifying enzymes, and chromatin remodeling complexes. Proteins containing a Su(var)3-9, enhancer-of-zeste, trithorax (SET) domain are well-established regulators of gene expression primarily through catalyzing methylation of lysine residues within histones (Jaiswal et al, 2017; Husmann & Gozani, 2019), although SET domain proteins also methylate non-histone substrates (Carlson et al, 2014; Jethmalani

& Green, 2020). A subfamily of SET domain proteins, often referred to as the Set3 subfamily, is characterized by divergent SET domains which appear to lack methyltransferase activity because of amino acid substitutions at key substrate binding interfaces (Dillon et al, 2005; Mas-Y-Mas et al, 2016). This subfamily includes the *Saccharomyces cerevisiae* paralogs Set3 and Set4, SET-9 and SET-26 from *Caenorhabditis elegans*, UpSET from *Drosophila melanogaster*, and the mammalian proteins MLL5 and SETD5 (Tran & Green, 2019b). Instead of directly catalyzing lysine methylation at chromatin, the Set3 subfamily of proteins are thought to regulate gene expression by binding to and regulating histone deacetylases (HDACs) at chromatin (Pijnappel et al, 2001; Kim & Buratowski, 2009).

Our previous work identified a role for the yeast protein Set4, a paralog to Set3, in protecting cells during oxidative stress, primarily through gene expression regulation of stress response genes (Tran et al, 2018). Set4 is lowly expressed in yeast cells under standard laboratory growth conditions, although deletion of *SET4* increases sensitivity to acute oxidative stress and alters gene expression patterns (Kemmeren et al, 2014; Tran et al, 2018; Tran & Green, 2019a, 2019b), indicating a biological function for Set4 even at low abundance. Expression of *SET4* appears to be stress-regulated, as the transcript and protein levels increase in low oxygen, including hypoxic or anaerobic conditions (Lai et al, 2006; Serratore et al, 2018). Other work has also implicated Set4 in the regulation of gene expression during hypoxia (Serratore et al, 2018), where it was shown to repress ergosterol biosynthetic genes together with the transcriptional repressor Hap1 through the inhibition of the sterol-responsive activator Upc2 (Serratore et al, 2018).

The subtelomeric regions of *S. cerevisiae* and other fungi, such as *Candida glabrata*, harbor many stress-response genes, particularly those that control adhesion, filamentation, and adaptation to anaerobic environments (Brown et al, 2010; De Las Peñas et al, 2015). Gene expression within subtelomeres is generally very low (Ellahi et al, 2015) because of regional silencing mechanisms, such as by the silent information regulator (SIR) histone deacetylase complex,

[1]Department of Biological Sciences, University of Maryland Baltimore County, Baltimore, MD, USA  [2]Department of Mathematics and Statistics, University of Maryland Baltimore County, Baltimore, MD, USA

Correspondence: egreen@umbc.edu
Yogita Jethmalani's present address is Stem Cell Translation Laboratory, National Center for Advancing Translational Sciences, National Institutes of Health, Rockville, MD, USA
Khoa Tran's present address is Laboratory of Cellular and Developmental Biology, National Institute of Diabetes and Digestive and Kidney Disorders, National Institutes of Health, Bethesda, MD, USA

▷▷▷ Life Science Alliance

and other transcriptional repressors (Jezek & Green, 2019). In other systems, Set3- and Set4-related proteins have been shown to maintain heterochromatic or repressive chromatin environments, including the fission yeast ortholog Set3 (Yu et al, 2016), the fly ortholog UpSET which interacts with the Rpd3/Sin3 deacetylase complex (Rincon-Arano et al, 2012; McElroy et al, 2017) and the *C. elegans* orthologs SET-9 and SET-26 which restrict spreading of H3K4me3-demarcated regions to regulate expression of germline-specific genes (Wang et al, 2018). Whereas Set3 in budding yeast is critical for gene repression in multiple contexts, it is not known to have a direct role in maintaining silent chromatin states such as at subtelomeres, the mating type locus, or ribosomal DNA locus (Pijnappel et al, 2001; Kim & Buratowski, 2009; Harvey et al, 2020). Given the structural similarities between yeast Set3 and Set4 and orthologous proteins (Tran & Green, 2019b), we hypothesized that Set4 may function in regulating silent chromatin regions in yeast, especially because genes required for multiple stress response pathways are found within silent regions such as subtelomeres (Jezek & Green, 2019). Here, we demonstrate that Set4 calibrates gene expression within yeast subtelomeres under both normal and stress conditions and contributes to cell fitness and cell wall integrity in hypoxic conditions. In hypoxia, Set4 promotes subtelomeric chromatin binding of the HDACs Sir2 and Rpd3, which have previously been implicated in the regulation of stress response and subtelomeric genes (Ai et al, 2002; Sertil et al, 2007; Alejandro-Osorio et al, 2009; Zhou et al, 2009; Ruiz-Roig et al, 2010; Radman-Livaja et al, 2011; Ellahi et al, 2015). This supports proper levels of histone acetylation and fine-tunes gene expression within the subtelomere. These data uncover a key function of Set4 in controlling subtelomeric chromatin to coordinate gene expression in response to stress. Furthermore, our results indicate that although this regulatory role of Set4 is performed under non-stress conditions, it becomes critical for cells in response to certain environmental signals, including oxidative stress and limiting oxygen.

## Results

### Subtelomeric gene expression is disrupted in *set4Δ* mutants

To better define the contribution of Set4 to gene expression and any potential roles in silent chromatin regulation, we performed an RNA-sequencing experiment of wild-type and *set4Δ* cells in un-stressed conditions (mid-log-phase growth, rich medium). Significantly differentially expressed genes were identified based on $\log_2$ fold-change (log FC) in *set4Δ* cells relative to wild type using local false discovery rate (FDR) ≤ 0.05 (see the Materials and Methods section; Table S1). In this analysis, 196 genes were identified as significantly differentially expressed in *set4Δ* cells, with 75 genes up-regulated and 121 genes down-regulated in the absence of Set4 (Fig 1A and Table 1). We performed gene ontology (GO) analysis to identify enriched categories of genes among those differentially expressed and identified no functional enrichment in genes down-regulated in *set4Δ* cells, although there is enrichment for genes involved in cell wall organization in those up-regulated in *set4Δ* cells (Table 1).

We noted that many of the genes associated with cell wall organization are encoded within subtelomeric regions (Ai et al, 2002). Therefore, we next assessed the enrichment of genes within 40 kb of chromosome ends to determine whether there is a more general enrichment for differential expression of subtelomeric genes in *set4Δ* cells that is independent of gene functional category. We observed a more than fivefold enrichment in genes adjacent to telomeres ($P = 1.60 \times 10^{-28}$ for all genes; hypergeometric test; Fig 1B) within those differentially expressed in *set4Δ* mutants. We also analyzed previously published microarray data of *set4Δ* cells grown in synthetic medium (Kemmeren et al, 2014). Interestingly, the differentially expressed genes in this dataset also showed significant enrichment for subtelomeric genes (*P*-value = 0.0003; Fig 1B), providing further evidence that Set4 may have a specific role in regulating expression of telomere-adjacent genes. In the same dataset (Kemmeren et al, 2014), gene expression in *set3Δ* cells was also analyzed, which showed no significant enrichment for differential expression of subtelomeric genes (*P*-value = 0.115). Together, these data suggest that under normal growth conditions, Set4 plays a specific role in regulating telomere-adjacent genes and genes linked to cell wall organization.

To further investigate these findings, we performed targeted gene expression analysis of wild-type and *set4Δ* cells using qRT-PCR. We observed that the mostly highly differentially expressed genes were associated with two common categories: (1) genes that are known targets of canonical silencing or telomere position effect (TPE) within subtelomeres; and (2), the seripauperin (*PAU*) genes, a highly homologous, subtelomeric gene family induced during different stresses—particularly anaerobic growth—that are thought to be important for cell wall remodeling or sterol uptake during stress (Rachidi et al, 2000; Luo & van Vuuren, 2009). In targeted qRT-PCR experiments, we analyzed expression of genes known to be regulated by TPE and chromatin-based silencing on the left arm of chromosome seven, *COS12* and *YGL262W*, as well as an adjacent gene *YPS5* (relative locations depicted in Fig 1C). Using primers that uniquely amplify these genes, we also monitored expression of *PAU11*, which is located adjacent to *TEL07L*, and *PAU13*, as well as *PAU21* and *PAU22*, which have identical sequences (indicated as *PAU21/22* where appropriate). In the absence of Set4, *COS12*, *YGL262W*, and *YPS5* were down-regulated, whereas *PAU11*, *PAU13*, and *PAU21/22* were up-regulated (Fig 1C). These data suggest Set4 is important for both maintaining expression of some subtelomeric genes and repressing other subtelomeric genes under physiological, un-stressed conditions.

The pattern of neighboring gene expression changes observed in *set4Δ* cells is consistent with a role for Set4 in altering regional chromatin structure. We therefore tested whether *set4Δ* cells showed any defects in a canonical TPE assay using a strain carrying *URA3* integrated near *TEL07L*. In this reporter assay, we did not observe any substantial change in *URA3* expression in the absence of Set4 or Set3, unlike the loss of silencing observed in *set1Δ* cells (Fig S1A). We also analyzed telomere length by Southern blot, which showed no difference between wild-type and *set4Δ* cells in the length of terminal telomere restriction fragments (Fig S1B). This indicates that Set4 has a specific regulatory role distinct from other TPE regulators and is not required for telomere length maintenance under normal conditions.

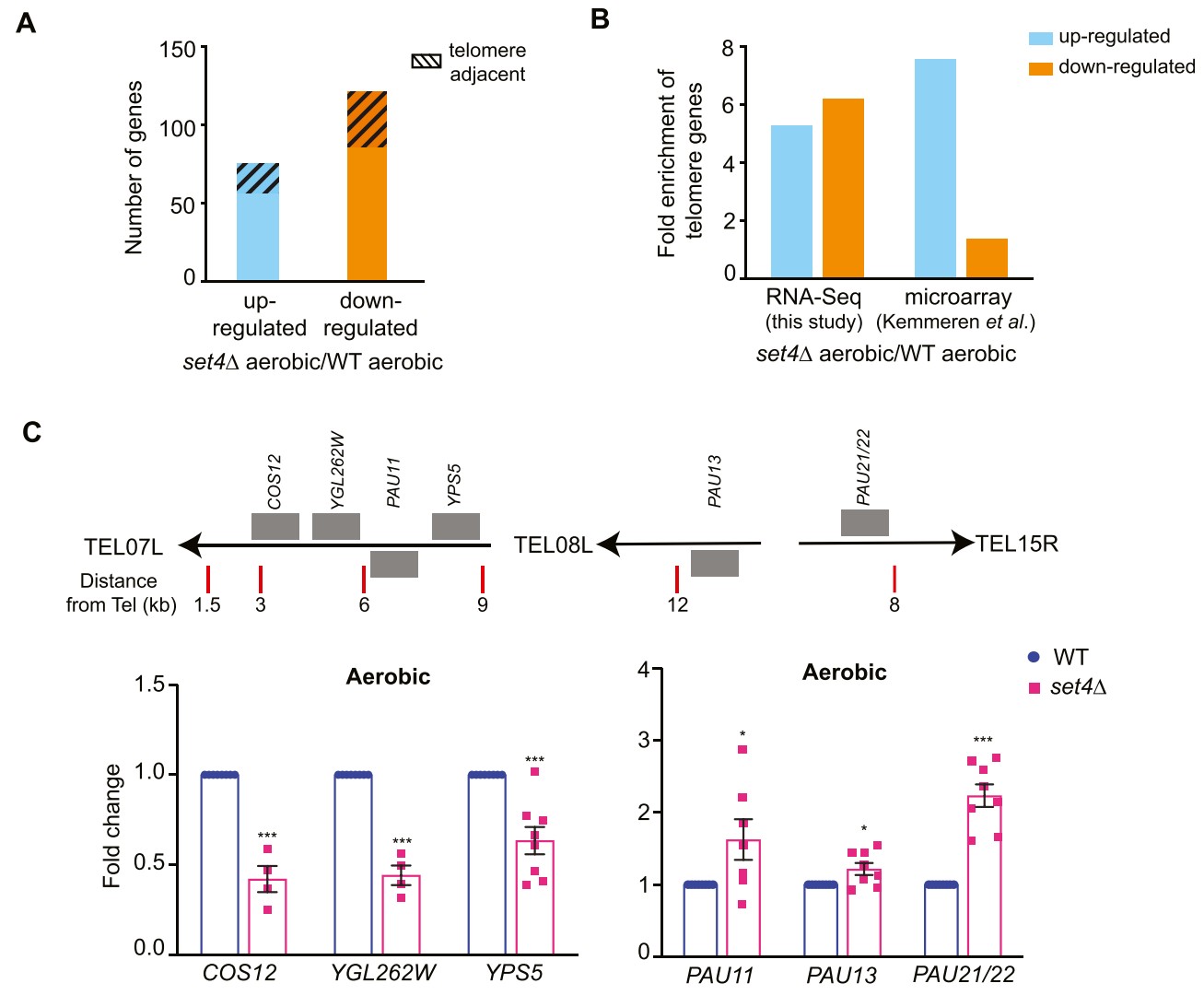

**Figure 1. Set4 regulates the expression of subtelomeric genes.**
**(A)** The total number of genes identified as up- or down-regulated (FDR ≤ 0.05) from RNA-sequencing of set4Δ (yEG322) cells relative to wt (yEG001). Gene list provided in Table S1. The total number of telomere-enriched genes is indicated with the hashed box. **(B)** The fold enrichment of differentially expressed subtelomeric genes (defined as less than 40 kb from the chromosome end) in our RNA-sequencing data of set4Δ cells and in previously published microarray data (Kemmeren et al, 2014). **(C)** qRT-PCR of sub-telomeric genes from wt (yEG001) and set4Δ (yEG322) strains grown in YPD. Expression levels were normalized to TFC1. Fold change relative to wt is shown. The error bars represent SEM from at least three biological replicates. Asterisks represent P-values as calculated by an unpaired t test (* ≤ 0.05; ** ≤ 0.01; *** ≤ 0.001).

## Deregulation of gene expression is enhanced in set4Δ mutants during stress

In previous work, we demonstrated that Set4 promotes proper gene expression in response to oxidative stress (Tran et al, 2018). Many genes encoded within subtelomeres are stress response genes; therefore we analyzed whether some of these genes showed Set4-dependent changes in expression during oxidative stress. Upon treatment with hydrogen peroxide, genes that were down-regulated in set4Δ cells did not show substantial change (COS12 and YGL262W; Fig S2A). Genes that were up-regulated in set4Δ cells under normal conditions (e.g., PAU13 and PAU21/22) were more highly up-regulated in set4Δ cells in the presence of hydrogen peroxide, although repressed in wild-type cells,

indicating that the loss of Set4 attenuates their repression in hydrogen peroxide.

Serratore et al (2018) previously showed that Set4 is important for the regulation of gene expression in hypoxic conditions and that hypoxia causes an increase in Set4 protein levels. As the PAU genes are highly up-regulated during hypoxia (Rachidi et al, 2000), we tested their expression in set4Δ cells, along with the other telomere genes, under hypoxic conditions. We first tested our growth conditions for wild-type cells grown in aerobic and hypoxic conditions. We obtained the most consistent results by diluting stationary phase cultures to a very low $OD_{600}$ and allowing them to grow to $OD_{600}$ ~0.4–0.8 over the course of 18 h in hypoxia, similar to how we tested set4Δ mutants sensitivity to oxidative stress (Tran et al, 2018; Tran & Green, 2019a). Under these conditions, the PAU genes were

**Table 1. Significant differentially expressed genes in *set4Δ* mutants compared with wild-type under aerobic and hypoxic conditions.**

| | Total genes | Up-regulated | Down-regulated |
|---|---|---|---|
| Aerobic | 196 | 75 | 121 |
| | | Cell wall organization ($9 \times 10^{-08}$) | No enrichment |
| Hypoxic | 377 | 205 | 172 |
| | | Cell wall organization ($3 \times 10^{-11}$) | Cell wall organization ($2 \times 10^{-04}$) |
| | | | DNA Integration ($2 \times 10^{-05}$) |

The number of significant differentially expressed genes in each category is indicated along with the enriched GO terms with *P*-values indicated in parentheses.

highly up-regulated, and there was also significant up-regulation of *YGL262W*, although expression of *COS12* and *YPS5* at *TEL07L* remained mostly unchanged in hypoxic compared with aerobic growth (Fig 2A). In the *set4Δ* strain, *COS12* and *YGL262W* expression showed no or minimal decrease in hypoxic conditions, whereas *PAU11*, *PAU13*, *PAU21/22*, and *YPS5* expression was significantly increased over the level of induction seen in wild-type cells (Fig 2B). These data indicate that the loss of Set4 leads to enhanced induction of hypoxia-regulated genes, including genes that are both negatively and positively regulated by Set4 under aerobic conditions (e.g., the *PAU* genes and *YPS5*, respectively). These observations parallel our findings in hydrogen peroxide treated cells, in which repression is inhibited at *PAU13*, *PAU21/22*, and *YPS5* (Fig S2A), indicative of a common gene regulatory role for Set4 under different stress conditions.

To address the role of Set4 in regulating subtelomeric genes more broadly during stress, we performed RNA-sequencing of wild-type and *set4Δ* cells grown under hypoxic conditions. In wild-type cells, growth in hypoxia induced widespread gene expression changes with 1,056 genes up-regulated and 835 genes down-regulated (log FC ≥ 1.0, *P* ≤ 0.05; Table S1 and Fig 2C). The significantly differentially expressed genes encompassed a range of GO categories, including enrichment for genes associated with transmembrane transport, lipid metabolic process, and cell wall organization, among others, in the up-regulated genes (Table S2). The genes down-regulated in wild-type cells in hypoxia were highly enriched for translation associated processes, mitotic cell cycle, cytoskeletal organization, cell wall organization, and lipid metabolic processes, among others (Table S2). The gene expression changes reported here are similar to those previously described under hypoxic or anaerobic growth of yeast (Kwast et al, 2002; Bendjilali et al, 2017).

In *set4Δ* cells, we observed a largely similar cohort of differentially expressed genes in hypoxia as in wild-type cells, with 1,073 genes up-regulated and 917 genes down-regulated (log FC ≥ 1.0, *P* ≤ 0.05; Table S1 and Fig 2D). These genes encompassed similar GO categories to those observed in wild-type cells (Table S2). There are more genes down-regulated in *set4Δ* cells grown in hypoxia compared to the total number of genes down-regulated in wild-type cells. These genes are distributed across a number of functional categories, including GO terms associated with translation-related processes, cytoskeletal organization, and DNA repair (Table S2).

When directly comparing wild-type and *set4Δ* cells in hypoxia, we identified 377 total genes differentially expressed, with 205 genes up-regulated and 172 genes down-regulated in the absence of Set4

(Table 1 and Fig 2E). GO analysis revealed enrichment for genes associated with cell wall organization in both the up- and down-regulated sets of genes and genes linked to DNA integration also enriched in the down-regulated genes (Table 1). Compared with aerobic conditions, this represents an increased number of cell wall organization genes misregulated in the absence of Set4. Interestingly, the down-regulated genes associated with the GO term DNA integration are almost entirely from Ty transposable elements (Table S1). Given that these are not differentially expressed under aerobic conditions, this indicates enhanced repression of these genes under hypoxia in *set4Δ* cells. In addition, previous work showed differential regulation of ergosterol biosynthetic genes in *set4Δ* mutants grown under hypoxia (Serratore et al, 2018). However, we did not observe enrichment of ergosterol biosynthetic genes within the differentially expressed gene set from our RNA-sequencing experiments (Table S1), nor was a large difference in expression observed in *ERG3* and *ERG11* using qRT-PCR (Fig S2B). It is possible that differences in yeast strains or growth conditions, such as the time in hypoxia, may contribute to this difference in expression patterns.

Based on results obtained under aerobic conditions and qRT-PCR experiments performed on telomere genes, we predicted that genes with altered expression in *set4Δ* cells in hypoxia may show subtelomeric enrichment. Indeed, for those genes up-regulated in hypoxic *set4Δ* cells, there was sixfold enrichment for subtelomeric localization compared with expected (*P* = $2.79 \times 10^{-31}$; hypergeometric test; Fig 2F) and almost twofold enrichment for subtelomeric localization for down-regulated genes (*P* = 0.008). These data support our conclusions from the qRT-PCR experiments indicating enhanced expression changes in cell wall organization genes at subtelomeres in *set4Δ* cells under hypoxia and indicate a broad role for Set4 in regulating subtelomeric genes under both normal and stress conditions.

## Set4 maintains cell wall integrity during hypoxic growth

Our previous work identified a role for Set4 in protecting cells during oxidative stress, likely through the regulation of gene expression. We showed that loss of Set4 increases sensitivity to oxidizing agents such as hydrogen peroxide, and Set4 overexpression increases survival upon hydrogen peroxide treatment (Tran et al, 2018). When growing cells under hypoxic conditions, we observed that *set4Δ* mutants grew more slowly and had smaller colony sizes than wild-type cells (Fig 3A), indicating impaired

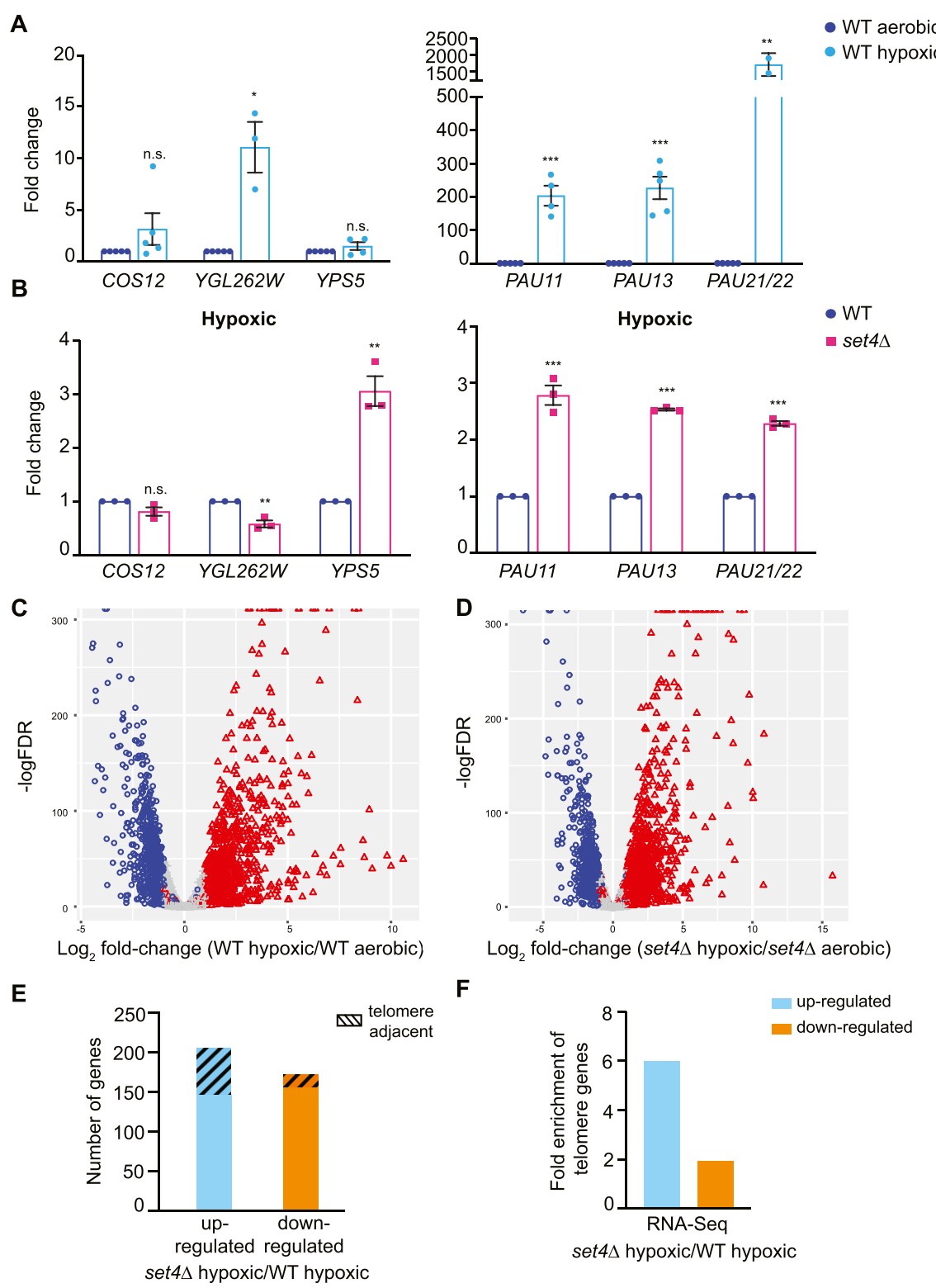

**Figure 2. Stress response genes at subtelomeres are regulated by Set4.**
**(A)** qRT-PCR of subtelomeric genes from wt (yEG001) strains grown in YPD under aerobic or hypoxic conditions. Expression levels were normalized to *TFC1* and fold change relative to aerobic conditions is shown. **(B)** qRT-PCR of subtelomeric genes from wt (yEG001) and *set4Δ* (yEG322) strains grown in hypoxia in YPD. Expression levels were normalized to *TFC1*. Fold change relative to wt in hypoxia is shown. For all panels, error bars represent SEM from at least three biological replicates and asterisks represent *P*-values as calculated by an unpaired *t* test (* ≤ 0.05; ** ≤ 0.01; *** ≤ 0.001; n.s., not significant). **(C, D)** Volcano plots depicting significantly differentially expressed genes (log FC ≥ 1.0, *P* ≤ 0.05) comparing wild-type hypoxic to wild-type aerobic cultures (C) and *set4Δ* hypoxic to *set4Δ* aerobic cultures (D). **(E)** The total number

growth under hypoxia. We hypothesized that the deregulated expression of the *PAU* genes, as well as other hypoxia-induced genes, may lead to disrupted cell wall integrity. We therefore tested the sensitivity of our strains to zymolyase digestion, which targets *β*-1,3 glucan linkages in the cell wall. As previously shown (Aguilar-Uscanga & François, 2003), yeast grown under hypoxic conditions showed increased resistance to zymolyase compared with aerobic growth (Fig 3B). However, *set4Δ* cells were modestly more sensitive to zymolyase digestion than wild-type cells in hypoxia, further indicating disrupted cell wall integrity.

Yeast cell walls show altered thickness and composition in hypoxia (Aguilar-Uscanga & François, 2003), and this can be visualized using trypan blue, which stains yeast glucans and chitin (Liesche et al, 2015). Under aerobic conditions, we observed similar average intensity of trypan blue staining in wild-type and *set4Δ* cells at the cell perimeter (Fig 3C and D). In wild-type cells in hypoxia, chitin composition and cell wall mass decrease, lowering trypan blue staining (Liesche et al, 2015). We observed an expected decrease in trypan blue staining in wild-type cells grown in hypoxia (Fig 3C and D), although staining has a higher mean intensity in *set4Δ* cells in hypoxia. This suggests that the cell wall remodeling typical of hypoxic cells is attenuated in *set4Δ* mutants. Our imaging analysis did not reveal a difference in cell size between the wild type and mutant under the different conditions (Fig 3D). Altogether, the increased zymolyase sensitivity and altered cell wall composition in *set4Δ* cells in hypoxia indicate a disrupted physiological response to stress in these mutants.

### Set4 maintains histone acetylation levels at stress response genes within subtelomeric regions

Our data show deregulation of hypoxia response genes in the absence of Set4, particularly those located within subtelomeric regions and important for cell wall integrity. Multiple histone deacetylases (HDACs) have been shown to control expression of stress response genes within subtelomeric regions and are key regulators of the repressive chromatin environment at subtelomeres (Ai et al, 2002; Sertil et al, 2007; Zhou et al, 2009; Radman-Livaja et al, 2011). Orthologs of Set4 in other organisms and the yeast protein Set3 are known to interact with or otherwise regulate the activity of HDACs in different chromatin contexts (Pijnappel et al, 2001; Rincon-Arano et al, 2012; Yu et al, 2016; Deliu et al, 2018; Tran & Green, 2019b; Wang et al, 2020). Thus, we hypothesized that Set4 may play a similar role at subtelomeric regions in yeast. To investigate this further, we tested the distribution of a series of acetylation marks previously implicated in the regulation of subtelomeric chromatin, including H4K5ac, H4K12ac, H4K16ac, and H3K9ac. We used primer sets that distinguished the repressed, silent chromosome end (*TEL07L*) and an internal site 12 kb from the chromosome end (*TEL07L$_{boundary}$*) which marks the approximate boundary with euchromatin that is adjacent to the

repressed subtelomeric genes *COS12* and *YGL262W*. We also tested the abundance of these modifications at the *PAU* gene promoters.

In aerobic conditions, we observed lower levels of histone acetylation close to the telomere (*TEL07L* primer set), and increased acetylation levels at more distal regions such as the *TEL07L* boundary region (Fig 4A). This is the expected distribution pattern of histone acetylation at subtelomeres and provides both positive and negative controls for the chromatin immunoprecipitation (chIP). In the absence of Set4, there was relatively little change in the abundance of these marks at any of the regions tested under aerobic growth. However, in hypoxic conditions, we observed increased acetylation, particularly at telomere-distal locations and the promoters of the *PAU* genes (Fig 4B). The largest increase in acetylation was observed for H3K9ac, although H4K16ac and H4K5ac also showed marked increases at subtelomeric regions in *set4Δ* cells. The overall abundance of histone acetyl marks was not changed in *set4Δ* cells, but we did observe a global decrease in H4K16ac in hypoxic conditions compared to aerobic conditions (Fig S3A). These findings demonstrate increased acetylation at multiple histone residues upon loss of Set4 in hypoxic conditions, consistent with our observations of enhanced activation of the *PAU* genes and less repression of other subtelomeric genes (e.g., *COS12* and *YGL262W*) in *set4Δ* cells.

Methylation of H3K4 by Set1 has also been linked to the regulation of subtelomeric gene expression (Nislow et al, 1997; Santos-Rosa et al, 2004; Margaritis et al, 2012; Jezek & Green, 2019). H3K4me3 is typically enriched at the boundary with euchromatin and is associated with promoters of highly transcribed genes (Kirmizis et al, 2007). chIP within the subtelomere and at *PAU* gene promoters showed no change in H3K4me3 levels in *set4Δ* cells in aerobic or hypoxic conditions (Fig 5A). This suggests that H3K4me3 is not regulated by Set4 at subtelomeric regions, nor does it appear to play a role in the regulation of *PAU* gene expression.

We also monitored the distribution of H3K36 methylation at subtelomeric regions and the *PAU* genes using chIP under aerobic and hypoxic conditions. H3K36me3 is linked to transcriptional elongation and the repression of cryptic transcription, particularly from internal promoters (Carrozza et al, 2005; Keogh et al, 2005), and is enriched within the coding regions of genes. We therefore included primer sets that anneal to *PAU* gene ORFs, in addition to promoter sequences, to test changes in H3K36me3 within the gene bodies. As expected, we observed lower levels of H3K36me3 at subtelomeric regions and *PAU* gene promoters, and higher levels within the ORFs (Fig 5B). This pattern was consistent under both aerobic and hypoxic conditions; however, there is an increase in H3K36me3 levels at *PAU* gene ORFs in hypoxia (compare y-axes in Fig 5B), despite no difference in bulk levels of H3K36me3 between the two conditions (Fig S3A). In *set4Δ* cells, we observed almost no change in H3K36me3 levels compared to wild-type (Fig 5B), suggesting that H3K36me3 is not affected by loss of Set4 in the regions tested, nor does it appear to play a major regulatory role in these regions. Consistent with these findings, cells without the H3K36

of genes identified as up- or down-regulated (FDR < 0.05) from RNA-sequencing of *set4Δ* (yEG322) cells relative to wt (yEG001) in hypoxia. Gene list provided in Table S1. The total number of telomere-enriched genes are indicated with the hashed box. **(F)** The fold enrichment of subtelomeric genes (defined as less than 40 kb from the chromosome end) for those genes differentially expressed between *set4Δ* hypoxic cultures relative to wild-type hypoxic cultures.

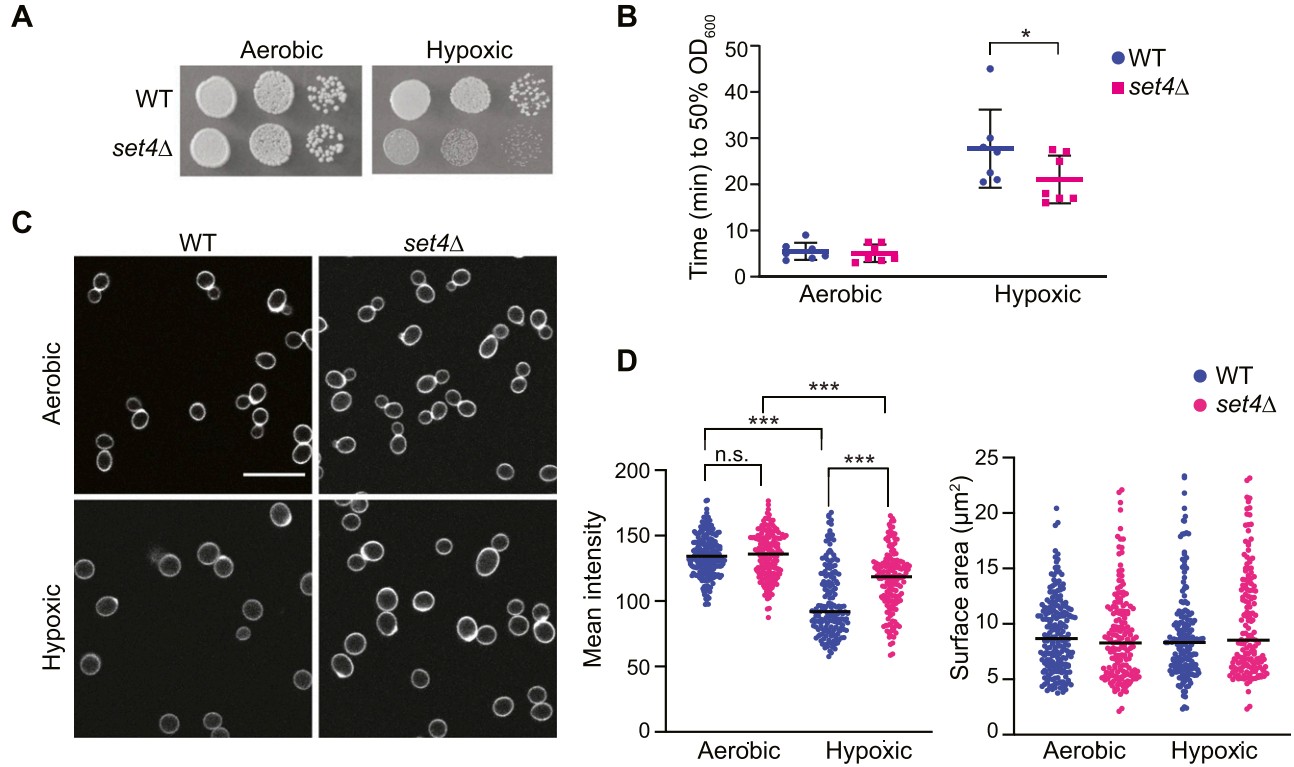

**Figure 3. Set4 promotes cell fitness and cell wall integrity in hypoxia.**
**(A)** Ten-fold serial dilutions of wt (yEG001) and set4Δ (yEG322) strains spotted on YPD and grown under aerobic (2 d) or hypoxic (8 d) conditions at 30°C. **(B)** Scatter dot plot of the time elapsed for wt (yEG001) and set4Δ (yEG322) cultures grown in either aerobic or hypoxic conditions to reach 50% digestion by zymolyase. Error bars represent SD from seven biological replicates. Asterisk represents P-value as calculated by two-way ANOVA and Sidak's multiple comparisons test (* <0.05). **(C)** Fluorescence microscopy of wt (yEG001) and set4Δ (yEG322) cells grown under aerobic or hypoxic conditions and stained with trypan blue. Scale bar is 5 $\mu$m. **(D)** Left panel: Quantitation of the mean intensity of the trypan blue staining at the cell perimeter of the images shown in (C). Black line indicates the mean. Measurements were performed for 160–210 cells per genotype and condition. Asterisks represent P-value as calculated from a two-way ANOVA with Turkey's post hoc test (*** ≤ 0.001; n.s., not significant). Right panel: Surface area ($\mu$m²) of cells displayed and analyzed in (C). No significant differences were found using a two-way ANOVA and Turkey's post hoc test.

methyltransferase Set2 showed decreased expression of *PAU* genes under both aerobic and hypoxic conditions (Fig S4) relative to wild type. As H3K36me3 and Set2 are primary repressors of cryptic transcription, these data suggest that the *PAU* genes and the other subtelomeric loci tested are not subject to high levels of cryptic transcription, and therefore Set4 is likely regulating expression through a different mechanism than the repression of cryptic transcripts from the *PAU* loci.

## Disrupted localization of HDACs at subtelomeric regions in *set4Δ* mutants

The HDACs Sir2 and Rpd3 are both known regulators of silent chromatin near telomeres (Zhou et al, 2009; Ehrenraut et al, 2010; Ellahi et al, 2015; Jezek & Green, 2019) and have also been implicated in the regulation of stress response genes, including those induced during hypoxic or anaerobic growth (Ai et al, 2002; Sertil et al, 2007; Radman-Livaja et al, 2011; Tung et al, 2013). These observations, and our findings of altered levels of histone acetylation in the absence of Set4, led us to test the hypothesis that Set4 works with HDACs to maintain telomeric chromatin structure. We investigated the distribution of Rpd3 and the SIR complex in wild-type and *set4Δ*

cells under hypoxic conditions. We focused on hypoxia as both the gene expression data and histone modification chIP data suggest a much larger dependence on Set4 in hypoxic than aerobic conditions.

The direct chromatin-interacting component of the SIR complex is Sir3, which serves to recruit the Sir2 HDAC and Sir4 to chromatin (Carmen et al, 2002; Liou et al, 2005; Altaf et al, 2007). We therefore performed chromatin immunoprecipitation of epitope-tagged Sir3 in *set4Δ* cells to assay subtelomeric binding of the SIR complex. We observed the expected occupancy of Sir3-HA primarily near telomeric chromatin at *TEL07L*, as well as secondary localization at the promoters of *PAU* genes, as previously demonstrated in aerobic conditions (Radman-Livaja et al, 2011). In hypoxia, Sir3-HA localization at telomeric chromatin decreased in *set4Δ* cells relative to wild type (Fig 6A), suggesting that Set4 promotes the proper association of the SIR complex with telomeres in these conditions. In agreement with previous findings (Radman-Livaja et al, 2011), we observed more binding of Sir3 to *PAU13* and *PAU11* promoters compared to *PAU21/22* promoters, indicating that *PAU13* and *PAU11* may be more dependent on the SIR complex for maintaining repression. These findings are also consistent with the increase in acetylation in the region observed in hypoxia (Fig 4), including

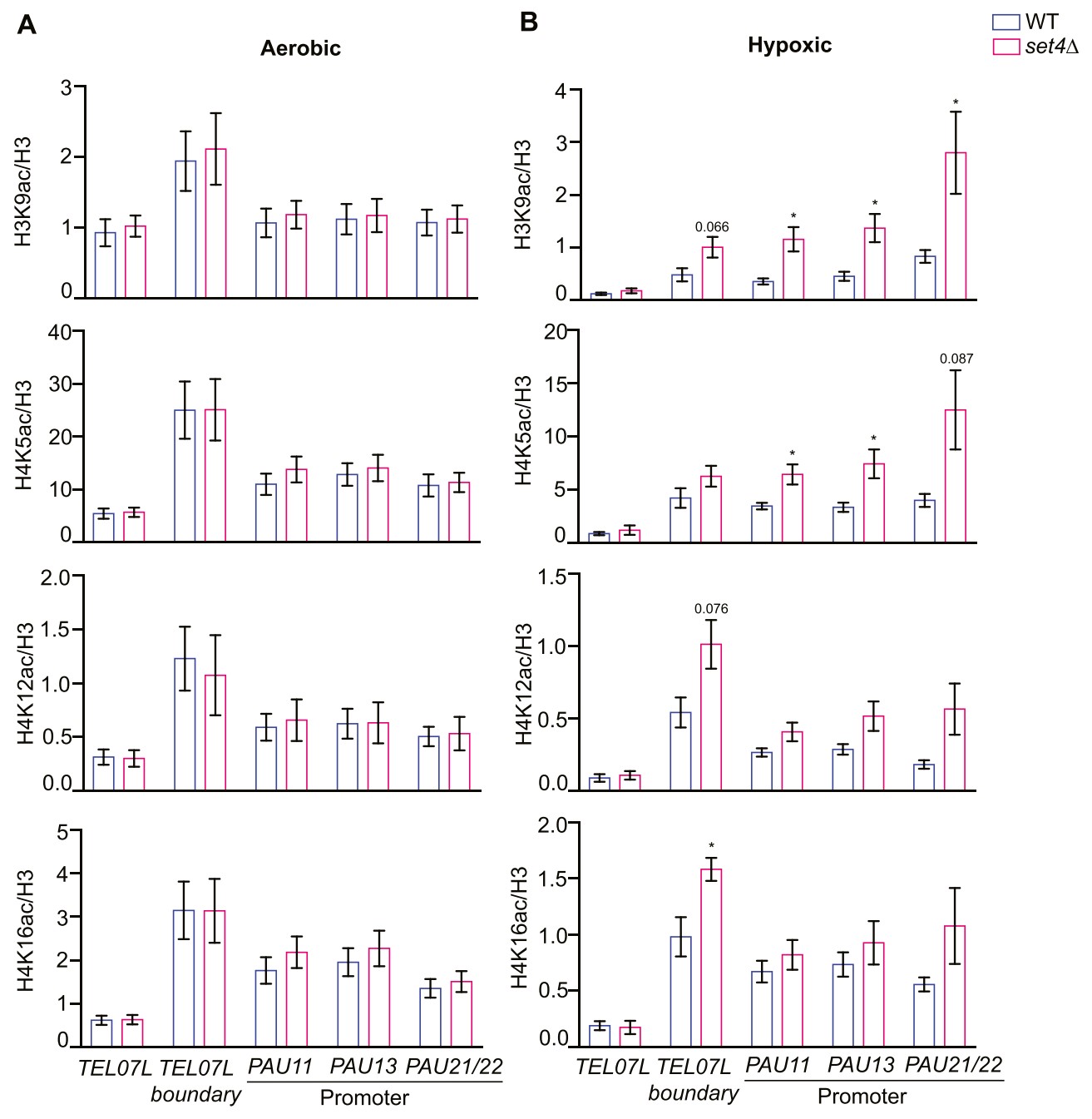

**Figure 4. Histone acetylation increases at subtelomeric chromatin in *set4Δ* cells in hypoxia.**
**(A, B)** chIP of H3K9ac, H4K5ac, H4K12ac, and H4K16ac at subtelomeric regions from wt (yEG001) and *set4Δ* (yEG322) strains grown to mid-log phase in YPD under aerobic (A) or hypoxic (B) conditions. Percent input of each acetyl mark is shown relative to percent input of total H3 levels. A minimum of three biological replicates for histone acetyl mark chIPs and six biological replicates of histone H3 chIP was performed. The histone H3 immunoprecipitation is more efficient and consistent than histone H4, and therefore was used to normalize to total histone levels. For all panels, error bars indicate SEM and asterisks represent *P*-values as calculated by unpaired t tests (* ≤ 0.05). If no asterisk is present, no significant differences were detected.

H4K16ac, the primary substrate of Sir2. We did not observe any differences in protein expression levels of Sir3-HA between wild-type and *set4Δ* cells (Fig S3B); however, there is increased chromatin binding by Sir3 in hypoxic cells (Fig S5A, compare y-axes of aerobic and hypoxic graphs). This observation is consistent with the global decrease in H4K16ac levels detected in hypoxia-treated cells (Fig S3A).

We similarly evaluated the distribution of Rpd3-FLAG at subtelomeres in *set4Δ* cells. In hypoxic conditions, Rpd3-FLAG showed decreased binding in the absence of Set4 (Fig 6B) indicating that Set4 promotes the localization of Rpd3-FLAG to subtelomeric regions. There were no changes in total Rpd3-FLAG protein levels in *set4Δ* mutants (Fig S3C). These observations are consistent with

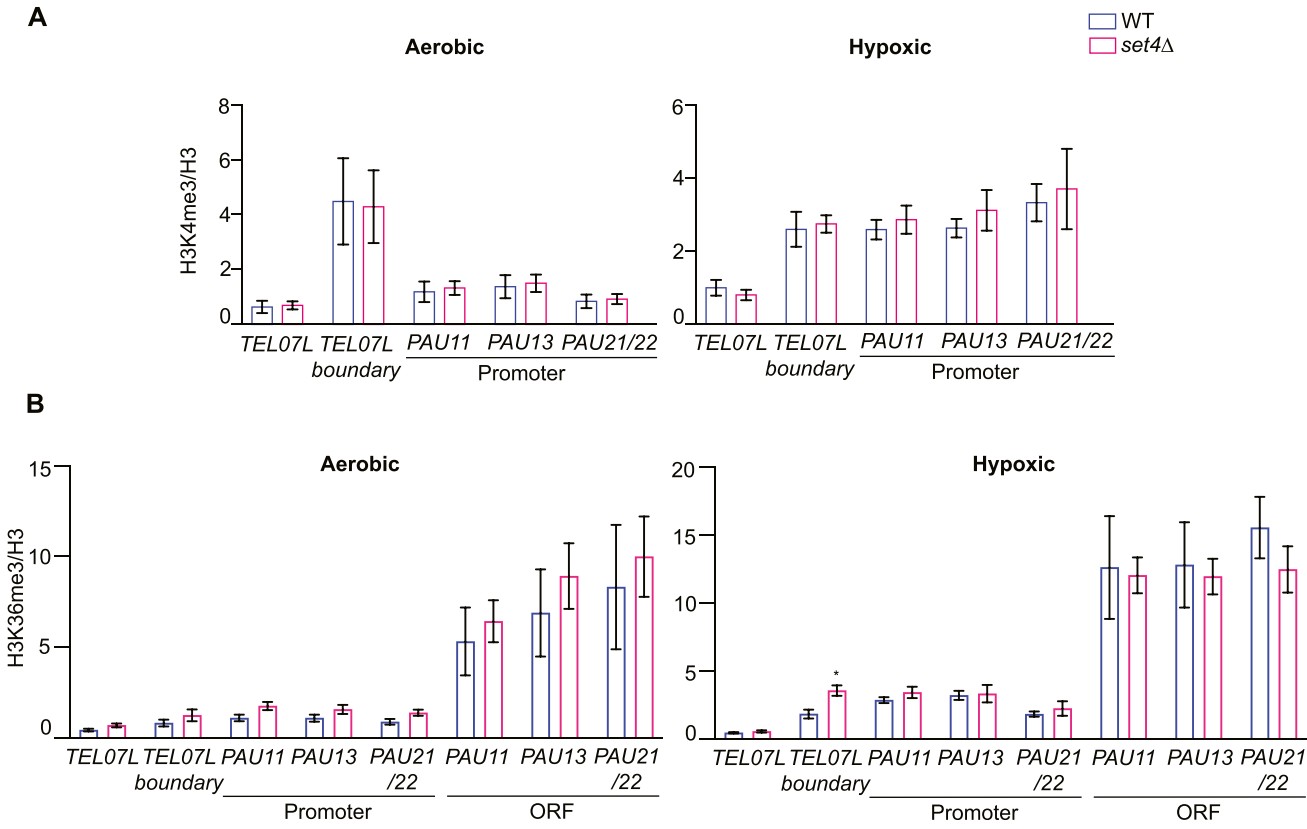

**Figure 5. H3K4me3 and H3K36me3 show little change in the absence of Set4 at subtelomeric regions.**
**(A, B)** chIP of H3K4me3 (A) and H3K36me3 (B) were performed as described and analyzed as presented in Fig 4. Because of the predominant localization of H3K36me3 in coding regions, additional primer sets within PAU gene ORFs were used for this chIP. Error bars indicate SEM and asterisks represent *P*-values as calculated by unpaired t tests (* ≤ 0.05). If no asterisk is present, no significant differences were detected.

increased acetylation within the region in *set4Δ* cells (Fig 4) and the altered gene expression patterns observed in *set4Δ* cells.

To further define the interaction of Set4 with the Sir2 and Rpd3 HDACs in regulating gene expression, we generated double mutant strains carrying *set4Δ* and *rpd3Δ* or *sir2Δ* and monitored gene expression changes and cell growth in hypoxia. Both *set4Δ* and *rpd3Δ* cells showed growth defects in hypoxic conditions compared with aerobic conditions, although there was no clear defect in *sir2Δ* cells and *sir2Δ set4Δ* cells grew very similarly to *set4Δ* single mutants (Fig 7A). These data are further quantified in Fig S5B. The loss of Rpd3 resulted in a severe growth defect in hypoxia (Figs 7A and S5B), and *rpd3Δ set4Δ* cells grew similarly to *rpd3Δ* single mutants, suggesting that Rpd3 and Set4 may contribute to a shared pathway regulating growth in hypoxia.

We next evaluated gene expression at subtelomeres in these single and double mutant strains. As expected, we observed de-repression of telomere-adjacent genes in *sir2Δ* cells under aerobic conditions (Fig S5C). In contrast, we observed enhanced repression of telomere-adjacent genes *COS12*, *YGL262W*, and *YPS5*, as well as lower expression of two of three *PAU* genes tested, *PAU11* and *PAU13*, in *rpd3Δ* cells (Fig S5C). These data are consistent with previous reports indicating an anti-silencing function for Rpd3 at subtelomeres (Zhou et al, 2009).

In *sir2Δ* cells grown in hypoxic conditions, *COS12* and *YGL262W* were de-repressed, as expected (Fig 7B). Similar expression was

observed in the *sir2Δ set4Δ* mutants, indicating that expression levels of these genes are largely regulated by the SIR complex in hypoxia. However, we observed a different expression pattern of genes that show enhanced activation in *set4Δ* cells in hypoxia, including *YPS5* and the *PAU* genes. These genes showed increased repression in *sir2Δ* cells compared with wild-type in hypoxia; however, this repression was alleviated in the *sir2Δ set4Δ* double mutants. *PAU11* and *PAU13* were expressed at a similar level in the double mutant as wild-type cells, whereas *PAU21/22* and *YPS5* showed slightly higher expression than in wild type. These data indicate an antagonistic function of the SIR complex and Set4 in balancing the expression of hypoxia-induced genes.

We also investigated changes in gene expression in the *rpd3Δ set4Δ* strain under hypoxic conditions. Rpd3 has been directly implicated in regulating the expression of genes induced during anaerobic growth (Sertil et al, 2007). In the absence of Rpd3, repression of the subtelomeric genes remained largely intact compared with the induction observed in wild-type and *set4Δ* cells in hypoxic conditions. However, in the *rpd3Δ set4Δ* cells, repression of the *PAU* genes was relieved and induction closer to wild-type expression levels was observed (Fig 7C). The telomere-adjacent genes *COS12* and *YGL262W* showed similar expression levels in *rpd3Δ* and *rpd3Δ set4Δ* cells, indicating that loss of Set4 was not sufficient to overcome repression of these genes in the absence of

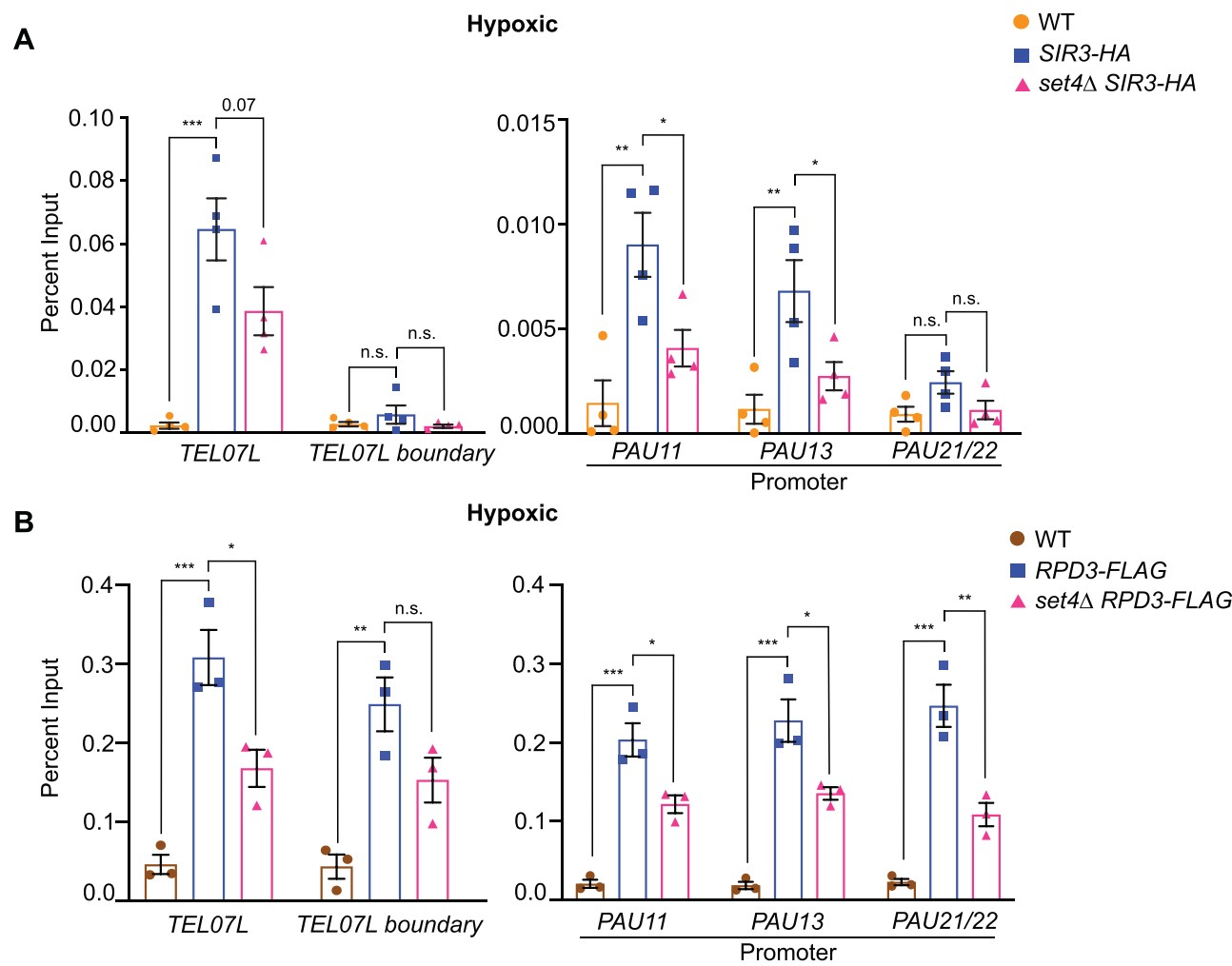

**Figure 6. Disrupted HDAC distribution at subtelomeric chromatin in the absence of Set4.**
**(A)** chIP of wt (yEG001), *SIR3-HA* (yEG873), and *SIR3-HA set4*Δ (yEG874) strains grown to mid-log phase in YPD in hypoxic conditions. **(B)** chIP of wt (yEG001), *RPD3-FLAG* (yEG956) and *RPD3-FLAG set4*Δ (yEG1010) strains grown to mid-log phase in YPD in hypoxic conditions. For both panels, percent input from at least three biological replicates is shown. The error bars indicate SEM and asterisks represent *P*-values as calculated by one-way ANOVA and Tukey's post hoc test (* ≤ 0.05; **≤ 0.01; *** ≤ 0.001; n.s., not significant).

Rpd3. These data suggest that, similar to its interaction with the SIR complex, Set4 counterbalances Rpd3 function in regulating expression of the *PAU* genes (and likely other genes induced in limiting oxygen). However, this is a gene-specific interaction, as Set4 and Rpd3 appear to function independently at other telomere adjacent genes such as *COS12*, *YGL262W*, and *YPS5*.

### The transcription factor Upc2 is required for increased activation of *PAU* genes in *set4*Δ cells

The primary transcription factor known to activate *PAU* gene expression in hypoxia is Upc2, which binds specific sequence elements upstream of most of the *PAU* genes (Hickman et al, 2011). Previous work showed that Set4 antagonizes Upc2 in regulating the expression of ergosterol biosynthetic genes in hypoxia (Serratore et al, 2018). To investigate the interaction of Set4 and Upc2 in *PAU* gene regulation, we generated *upc2*Δ and *upc2*Δ *set4*Δ yeast strains

and tested expression of *PAU* genes and other subtelomeric loci in hypoxia. We observed a clear requirement for Upc2 in activating expression of the *PAU* genes in hypoxia, as almost no *PAU* expression was found in *upc2*Δ cells (Fig 8). In *upc2*Δ *set4*Δ cells, there was also no induction of *PAU* gene expression. This suggests that the enhanced activation of *PAU* genes in *set4*Δ cells depends on the presence of Upc2. In addition, we noted that the other, non-*PAU* subtelomeric genes showed expression levels similar to wild-type in the *upc2*Δ *set4*Δ cells. This suggests that changes in subtelomeric chromatin in *set4*Δ cells that influence expression of *COS12*, *YGL262W*, and *YPS5* may also depend on Upc2 or the activation of neighboring *PAU* gene transcription.

### Set4 localizes to subtelomeric chromatin in hypoxia

Previous work from our laboratory showed that Set4 is a chromatin-associated protein and localizes to the promoters of genes that are

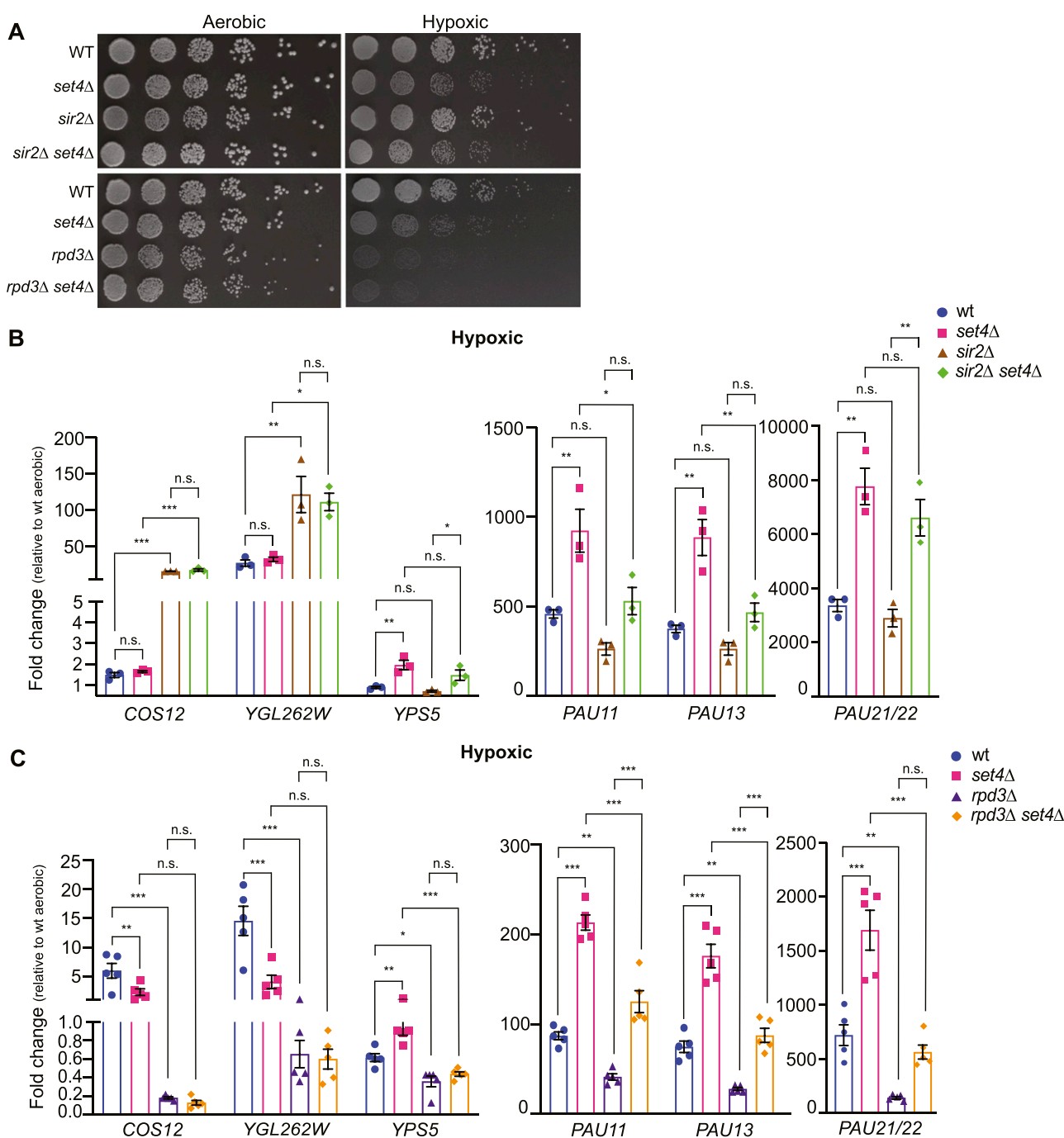

**Figure 7. Genetic interactions of Set4 with HDACs Sir2 and Rpd3 in regulating subtelomeric gene expression during stress.**
**(A)** Ten-fold serial dilutions of wt (yEG001), set4Δ (yEG322), sir2Δ (yEG917), sir2Δ set4Δ (yEG997), and wt (yEG919), set4Δ (yEG920), rpd3Δ (yEG921), rpd3Δ set4Δ (yEG922) spotted on YPD and grown in aerobic or hypoxic conditions. Images of aerobic plates were taken after 2 d of growth at 30°C and images of hypoxic plates were taken after 8 d of growth. **(B)** qRT-PCR of subtelomeric genes from wt, set4Δ, sir2Δ, and set4Δ sir2Δ strains grown under hypoxic conditions in YPD. **(C)** qRT-PCR of subtelomeric genes from wt, set4Δ, rpd3Δ, and set4Δ rpd3Δ strains grown under hypoxic conditions in YPD. For all experiments, expression levels were normalized to *TFC1* and fold-change was determined relative to wild-type expression levels in aerobic conditions. Error bars represent SEM from at least three biological replicates. For all panels, asterisks represent *P*-values as calculated by one-way ANOVA and Tukey's post hoc test (* ≤ 0.05; **≤ 0.01; *** ≤ 0.001; n.s., not significant).

induced during oxidative stress, particularly in the presence of stress (Tran et al, 2018). Another report has also shown that Set4 localizes to promoters of ergosterol biosynthetic genes during hypoxia (Serratore et al, 2018). To investigate the localization of Set4

at subtelomeres and whether the changes in subtelomeric gene expression and acetylation levels are due to local occupancy by Set4, we performed chIP under both aerobic and hypoxic conditions using a strain expressing N-terminally FLAG-tagged Set4

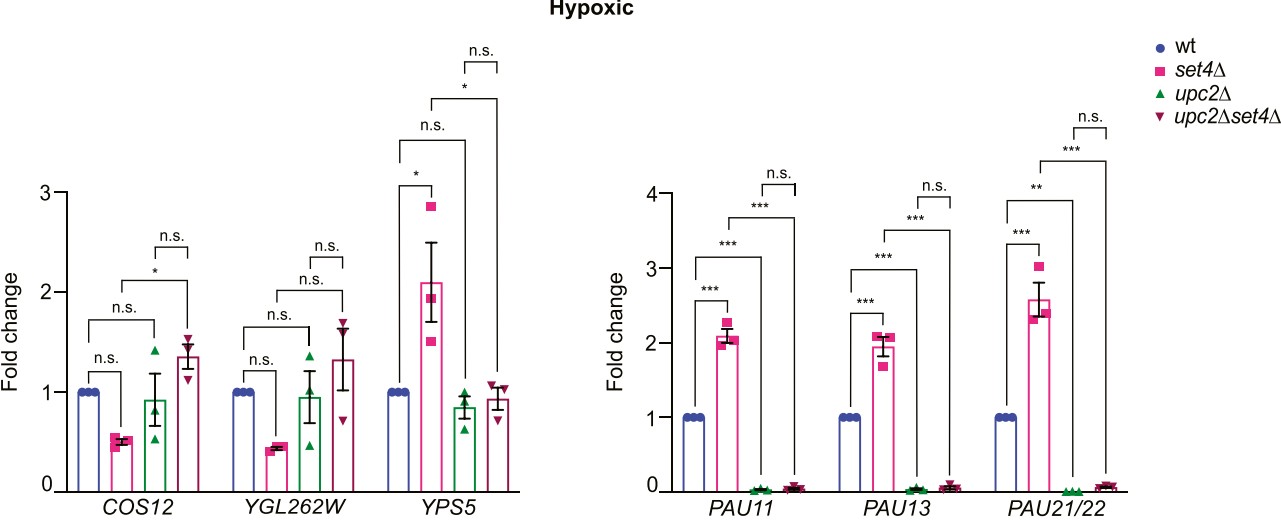

**Figure 8. Upc2 is required for enhanced activation of *PAU* genes in the absence of Set4.**
qRT-PCR of subtelomeric genes from wt, *set4Δ*, *upc2Δ*, and *set4Δ upc2Δ* strains grown under hypoxic conditions in YPD. Expression levels were normalized to *TFC1* and fold-change was determined relative to wild-type expression levels in hypoxic conditions. Error bars represent SEM from three biological replicates. Asterisks represent *P*-values as calculated by one-way ANOVA and Tukey's post hoc test (* ≤ 0.05; **≤ 0.01; *** ≤ 0.001; n.s., not significant).

from its endogenous locus and monitored binding to *TEL07L* and *TEL07L_{boundary}* regions and the promoters of *PAU* genes. In aerobic conditions, we did not detect significant association of Set4 at any of these regions (Fig S6). However, under hypoxia, we observed binding of Set4 to a region near *TEL07L* as well as the promoters of the *PAU* genes (Fig 9). This binding was enhanced relative to negative control regions, *CENXV* and the gene *PRP8*, which showed no change in gene expression in *set4Δ* cells (Table S1). In addition, we also tested the promoters of other non-telomeric genes known to be regulated by Set4 during stress

(*CTT1*, *PNC1*, *ERG3*, and *ERG11*) (Serratore et al, 2018; Tran et al, 2018). Set4 localized to the promoters of *CTT1* and *PNC1*, as expected based on our previous findings (Tran et al, 2018), and was highly enriched at *ERG3* and *ERG11* gene promoter (Fig 9). These results are consistent with a previous report showing binding to *ERG3* and *ERG11* promoters in hypoxia (Serratore et al, 2018); however, we did not observe these to be major sites of gene regulation by Set4 under similar conditions (Fig S2B).

The binding of Set4 to subtelomeric chromatin under stress suggests that Set4 may directly influence gene expression within

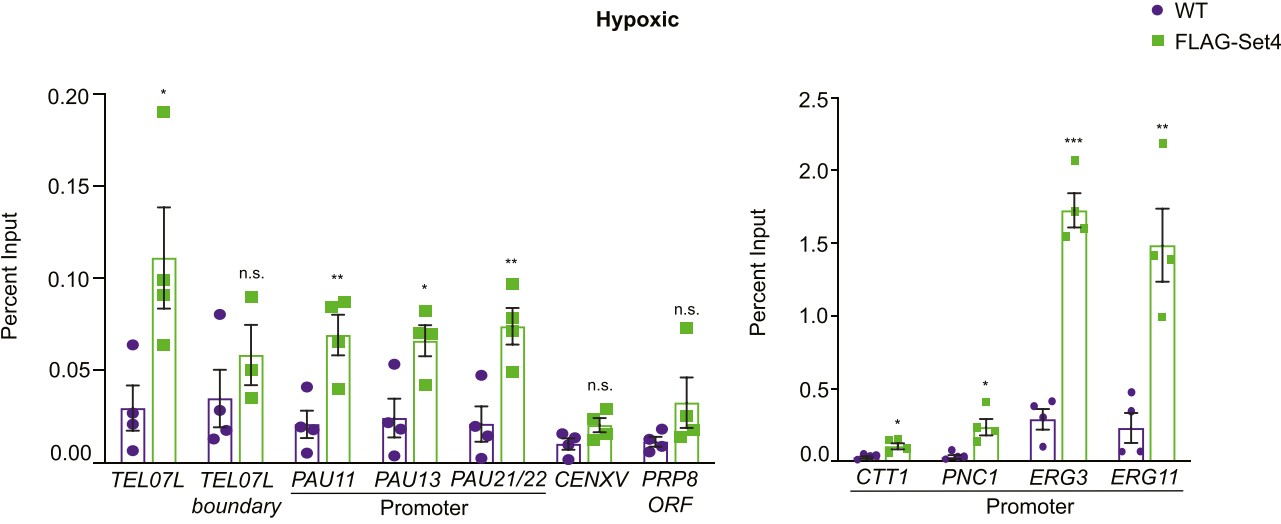

**Figure 9. Set4 localizes to subtelomeric chromatin during stress.**
chIP of FLAG-Set4 from cells grown under hypoxic conditions. Percent input from three biological replicates is shown. Left graph shows regions tested in Figs 4 and 6 for histone acetylation levels and HDAC binding, as well as the negative control regions *CENXV* and the *PRP8* ORF. Right graph shows promoter regions previously identified as binding locations under stress conditions (Serratore et al, 2018; Tran et al, 2018). Error bars represent SEM and asterisks represent *P*-values as calculated by an unpaired *t* test (* ≤ 0.05; **≤ 0.01; *** ≤ 0.001; n.s., not significant).

the region. Set4 expression increases dramatically during hypoxia (Serratore et al, 2018); therefore, we expect increased association with chromatin under these conditions. In aerobic conditions, the abundance of Set4 is very low (Tran et al, 2018), which likely limits our ability to detect it by chIP. Combined with our gene expression data, we expect that Set4 may be present at subtelomeric chromatin at levels below the limit of detection in aerobic conditions, and Set4 abundance and localization near telomeres increases in hypoxia.

# Discussion

Yeast subtelomeres are enriched for stress response genes, and proteins orthologous to Set4 are known regulators of heterochromatin and gene silencing (Rincon-Arano et al, 2012; Yu et al, 2016; McElroy et al, 2017; Wang et al, 2018). Previous studies have highlighted a role for Set4 as a calibrator of stress-responsive gene expression (Serratore et al, 2018; Tran et al, 2018). Here, we uncovered a function for Set4 in regulating genes within the repressed subtelomeric regions of budding yeast under both normal and stress conditions, particularly during hypoxia. Gene expression and chromatin immunoprecipitation analysis indicate that Set4 works together with the SIR complex and Rpd3 within the subtelomeres to fine-tune expression levels of stress response genes. The loss of Set4 also decreases survival and cell wall integrity in hypoxia. Therefore, Set4 helps to maintain the proper balance of expression of stress response genes to promote survival during stress.

### Set4-dependent regulation of subtelomeric gene expression under both normal and stress conditions

Previous work has shown that Set4 localizes to the promoters of oxidative stress-induced genes after hydrogen peroxide treatment (Tran et al, 2018) and ergosterol biosynthetic genes during hypoxia (Serratore et al, 2018). Set4 is lowly expressed under normal conditions, and its localization to these promoters was only detected during stress. We also observed enrichment of Set4 within subtelomeric regions, specifically during hypoxia, when Set4 protein abundance is dramatically increased (Serratore et al, 2018). Changes in gene expression of telomere-adjacent genes and the stress-induced *PAU* genes were observed under both normal and stress conditions; however, the dependence on Set4 was clearly enhanced during stress. RNA-seq revealed an overall increase in differential gene expression between wild-type and set4Δ cells in hypoxia, although we note that many of the gene expression changes observed in hypoxia also occurred under aerobic conditions, though the magnitude of the differences between wild-type and set4Δ cells was lower. This is most apparent from our analysis of GO categories (Table 1) and subtelomeric enrichment (Figs 1A and B and 2E and F), which both identified common categories of genes differentially expressed in set4Δ cells under aerobic and hypoxic conditions.

Consistent with changes in gene expression, there were greater changes in histone acetylation levels in hypoxia compared with normal conditions in set4Δ cells. We postulate that Set4 is present

within subtelomeres (and likely other chromatin regions) even under normal, unstressed conditions, as we observe Set4-dependent changes in gene expression; however, the standard chIP assay used is not sufficiently sensitive to detect this low abundance protein. In hypoxic conditions, the differences in gene expression and histone acetylation in set4Δ cells compared with wild-type cells are exacerbated, and we observe a clear localization of Set4 to subtelomeric chromatin. The increased abundance of Set4 in hypoxia (Serratore et al, 2018) allows us to readily detect the protein using chIP. Combined with our previous results showing increased chromatin association of Set4 during oxidative stress (Tran et al, 2018), these data indicate that the gene regulatory role for Set4 is more critical during stress. This suggests that one component of the cellular response to certain types of stress is to increase Set4 protein levels and/or increase its association with chromatin to promote stress-responsive gene expression programs. Currently, this role for Set4 has only been linked to oxidative stress and limiting oxygen (hypoxic or anaerobic) conditions. It remains to be determined whether or not Set4 is a general stress response factor, similar to the Msn2 and Msn4 transcription factors (Morano et al, 2012), or if it has a specialized role under certain types of stress.

### Set4 coordinates histone deacetylases to regulate subtelomeric chromatin structure

The chromatin structure at subtelomeric regions of *S. cerevisiae* is maintained by multiple HDACs to generate a hypoacetylated state, which keeps gene expression levels low (Ellahi et al, 2015; Jezek & Green, 2019). Members of the Set3 subfamily of SET-domain proteins, including Set3, UpSET, and SETD5 are all known to physically interact with histone deacetylases (Tran & Green, 2019b), and loss of function of these proteins leads to aberrantly high levels of histone acetylation (Kim & Buratowski, 2009; Rincon-Arano et al, 2012; Wang et al, 2020). Protein–protein interaction analysis under hypoxic conditions revealed interactions of Set4 with other chromatin regulators, although not HDAC complex members (Serratore et al, 2018); however, further analysis may reveal how Set4 influences HDAC function. Using chIP, we observed decreased binding of both the SIR complex and Rpd3 within subtelomeric regions in cells lacking Set4 under hypoxic conditions, when Set4 expression is high. Not all Rpd3 or SIR protein binding is lost in the absence of Set4, suggesting other targeting mechanisms of both HDACs are still intact. However, the reduced presence of each of these HDACs is consistent with increases in local histone acetylation in set4Δ cells in hypoxia (Fig 10). In addition, gene expression analysis in sir2Δ set4Δ and rpd3Δ set4Δ double mutants indicated that the repression of the *PAU* genes observed in the absence of either HDAC alone is relieved upon loss of Set4. At subtelomeric chromatin, Rpd3 has been reported to antagonize the spread of the SIR complex in silent chromatin regions (Zhou et al, 2009; Ehrentraut et al, 2010; Ellahi et al, 2015). Our data suggest a model in which the loss of Set4 diminishes the ability of either Sir2 or Rpd3 to fine-tune *PAU* gene expression levels, causing aberrantly high activation of these genes. Despite the antagonism between the SIR complex and Rpd3, the reduction in both of their levels in set4Δ mutants likely reduces

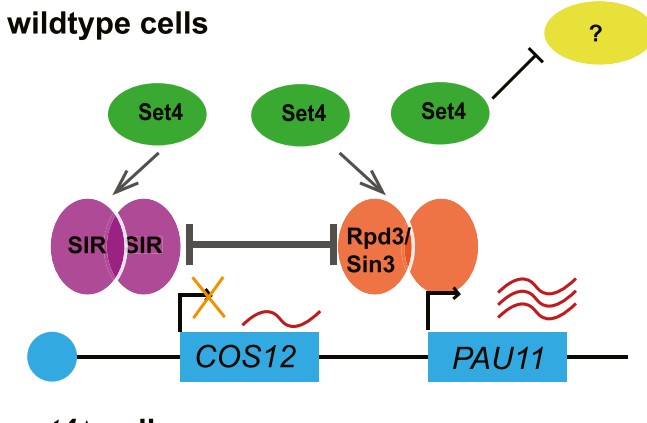

**wildtype cells**

**set4Δ cells**

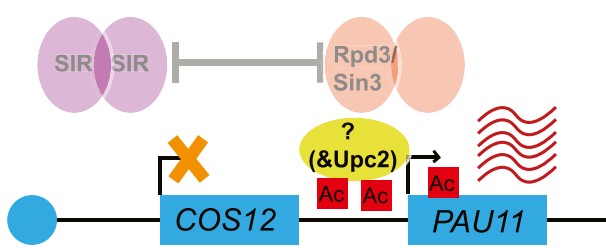

**Figure 10. Model for Set4 function in subtelomeric gene regulation during hypoxic stress.**
A partial depiction of *TEL07L* is shown indicating genes subject to telomere position effect (TPE) silencing, such as *COS12*, and genes repressed under standard growth conditions and induced in stress, such as *PAU11*. In wild-type cells, Set4 promotes the association of the SIR complex (Sir2/3/4) and Rpd3 with subtelomeric chromatin. The presence of these HDACs represses telomere-adjacent genes subject to TPE, such as *COS12*, and genes induced in limiting oxygen, such as *PAU11*. Set4, either alone or in cooperation with the SIR complex, Rpd3, or other yet unidentified chromatin regulators, may also inhibit the binding or activity of factors important for the positive regulation of stress response genes at subtelomeres (indicated by a question mark), including the hypoxic-responsive transcriptional activator Upc2. In the absence of Set4, both the SIR complex and Rpd3 binding are diminished, resulting in increased histone acetylation and enhanced activation of *PAU11* (and other *PAU*) genes. Genes subject to TPE, such as *COS12*, show increased repression upon loss of Set4, possibly because of diminishment of the antagonism between Rpd3 and the SIR complex, or because of compensation by other HDACs when Rpd3 and Sir2 levels are disrupted (Thurtle-Schmidt et al, 2016). This role for Set4 is most critical during stress, such as hypoxia when Set4 levels increase. Further genetic and physical interaction studies of Set4 at chromatin are likely to define the additional factors functioning with Set4, Rpd3, and the SIR complex in fine-tuning stress response genes within yeast subtelomeres.

improper spreading of the HDACs, allowing *PAU* gene expression to approach or exceed wild-type levels in *set4Δ sir2Δ* and *set4Δ rpd3Δ* mutants.

It is also feasible that, in addition to maintaining proper SIR complex and Rpd3 levels at subtelomeres, Set4 works alone or in cooperation with these HDACs to inhibit association of a positive regulator of hypoxia-induced genes. Previous work has demonstrated that Rpd3 promotes the association of the transcription factor Upc2 with some anaerobic response genes (Sertil et al, 2007). Upc2 is required for the induction of the *PAU* genes and works with the SAGA transcriptional activator and histone acetyltransferase complex to promote *PAU* gene expression in hypoxia (Hickman et al,

2011). Interestingly, it has been reported that Set4 antagonizes Upc2 activity at ergosterol biosynthetic genes, thereby repressing these genes in hypoxia (Serratore et al, 2018). We observed that deletion of Upc2 in *set4Δ* mutants eliminated all induction of the *PAU* genes, indicating that enhanced activation of these genes in *set4Δ* cells required Upc2 and that Set4's role in chromatin regulation is upstream of Upc2 at the *PAU* genes. Interestingly, other genes at the subtelomere (e.g., *COS12* and *YGL262W*) reverted to wild-type expression levels in *upc2Δ set4Δ*, suggesting that over-activation of the *PAU* genes in the absence of Set4 may cause misregulation of local chromatin structure and influence expression at nearby genes. Further investigation of the interaction of Set4 with Upc2, Rpd3, and Sir2 at subtelomeric regions will shed more light on regulatory mechanisms controlling their expression in hypoxia.

Altogether, this study provides new insights into the types of genes regulated by Set4 and the chromatin-based mechanisms through which it acts, as well as identifies a new telomere regulator in stress conditions. We have identified a role for Set4 in maintaining heterochromatic structures in yeast, which aligns its functions with metazoan orthologs previously implicated in heterochromatin maintenance (Rincon-Arano et al, 2012; McElroy et al, 2017; Wang et al, 2018), and expands our understanding of the role for Set4 during stress. Our data indicating decreased fitness and cell wall integrity of cells lacking Set4 in hypoxic conditions support the conclusion that Set4 promotes cell survival during stress, which is consistent with our previous findings identifying a role for Set4 in protecting cells during oxidative stress (Tran et al, 2018). Additional studies of Set4, and other Set3-related proteins, are likely to further our understanding of gene regulatory mechanisms and chromatin-mediated stress defense pathways.

## Materials and Methods

### Yeast strains and growth conditions

The genotypes for all *S. cerevisiae* strains used in this study are listed in Table S3. Strains carrying gene deletions were made using targeted PCR cassettes amplified from the pFA6a vector series (Longtine et al, 1998). Double mutant strains were isolated after haploid mating, sporulation, and tetrad dissection. All strain genotypes were confirmed by growth on the appropriate selective media and colony PCR using primers specific to individual gene deletions or epitope tag insertions. Standard media conditions for rich media (YPD; 1% yeast extract, 1% peptone, 2% dextrose) and synthetic complete (SC) or dropout media (US Biological) were used as necessary. For all growth assays, gene expression and chromatin immunoprecipitation experiments, yeast cultures were diluted and grown in appropriate media overnight to mid-log phase (OD$_{600}$ ~0.4–0.8) at 30°C. For hypoxic growth, the culture flasks were placed in BD GasPak EZ anaerobe pouch system and incubated at 30°C. For hydrogen peroxide–treated cultures, cells were grown to mid-log phase (OD$_{600}$ ~0.6–0.8) and then treated with 0.4 mM H$_2$O$_2$ for 30 min (Tran & Green, 2019a).

### RNA sequencing

Total RNA was extracted from yeast cells and subjected to Illumina-based RNA-sequencing at Genewiz. Differential gene expression

analysis was performed as previously described (Martín et al, 2014; Jezek et al, 2017a). Briefly, read quality control was analyzed using FastQC and adaptor removal and read trimming were performed with Trimommatic v.0.36 (Bolger et al, 2014). Reads were mapped to the *S. cerevisiae* reference genome using the STAR aligner v.2.5.2b and Subread package v.1.5.2 was used for calculating gene hit counts (Dobin et al, 2013). The data were normalized and log-fold change values were determined using DESeq2 (Huber et al, 2015). The raw and processed data for RNA-sequencing experiments are available on the Gene Expression Omnibus database at accession number GSE173901.

**Differential gene expression significance testing**

For testing the significance of gene expression changes, we used a hybrid of two existing methods depending on the applicability of zero assumption in Efron (2004): the center of the observed $\log_2$ fold-change (log FC) values consists of non-differentially expressed genes. One method is the local FDR procedure which estimates the distribution of the non-differentially expressed genes based on zero assumptions instead of using the standard normal distribution. It can be a more powerful test when zero assumption holds: log FC values showed little change at most genes with small groups of up- and down-regulated genes exhibiting the most change (Fig S7). Local FDR analysis was used to identify differentially expressed genes at FDR ≤ 0.05 for datasets comparing wild-type and *set4Δ* cells. The FDR is computed from these estimates and is controlled to be less than 5%. The method is implemented through the locfdr package in R (Efron et al, 2015).

The other method used for datasets comparing expression differences between aerobic and hypoxic conditions is the test procedure in DESeq2 after filtering absolute value of log FC > 1, the Wald test *P*-values which are adjusted (padjust) for multiple testing using the procedure of Benjamini and Hochberg (Benjamini & Hochberg, 1995). Because the test in DESeq2 does not need the structure assumption such as zero assumption, we applied it when there was large variability of log FC values and more of the non-differentially expressed genes spread out because of the discrepancy resulting from a larger number of up- and down-regulated genes. Padjust ≤ 0.05 are selected to be differentially expressed genes which represents FDR ≤ 0.05.

GO analysis was performed using the GO term function in Yeastmine and the GO term slim mapper through the Saccharomyces Genome Database. Telomere enrichment was determined by identifying the number of genes within 40 kb of the telomere end in each dataset analyzed and using a hypergeometric test to determine significance of enrichment and fold-enrichment over expected based on the total number of genes within subtelomeres in the genome.

**Gene expression analysis by quantitative reverse transcriptase PCR**

Total RNA was extracted using 1.5 ml of mid log phase culture of yeast cells ($OD_{600}$ ~0.6–0.8) under different growth conditions. Masterpure Yeast RNA purification kit (Epicentre) was used to extract the RNA by following the manufacturer's instructions. Turbo DNA-free kit (Ambion) was used to eliminate genomic DNA from the samples. cDNA was generated from 1 $\mu$g of total RNA using qMax cDNA synthesis kit (Accuris) containing both oligo dT and random hexamers for priming reverse transcription of RNA. For quantitative PCR (qPCR) to check transcript levels, 0.5 $\mu$l of cDNA was added to 1X qMax Green Low ROX qPCR mix (Accuris) with the appropriate gene specific primers (Table S4) in a 10-$\mu$l reaction. Real-time amplification was performed on a Bio-Rad CFX384 Real-time Detection System. Three technical replicates were performed for each reaction and a minimum of three biological replicates was performed for each experiment. Relative gene expression values were normalized to the control gene *TFC1*, whose expression has been shown to be stable under different growth conditions (Teste et al, 2009).

**Spot assays**

For the TPE spot assay, strains integrated with the *URA3* gene at *TELV07L* were used (see Table S3; kindly provided by Paul Kaufman). Gene knockouts were created using insertion of targeted PCR cassettes amplified from the pFA6a vector series (Longtine et al, 1998). Cells were grown overnight in YPD medium at 30°C and 0.1 OD units of the cultures were serially diluted and spotted on YPD plates (control) and 5-fluoroorotic acid (5-FOA) plates. The plates were observed and imaged for 2 d to analyze the growth pattern. For growth analysis of single and double mutant strains under aerobic and hypoxic conditions, yeast strains were grown overnight in YPD, diluted to $OD_{600}$ ~ 0.2 the next day, and grown to log phase. 0.1 $OD_{600}$ units of the culture were serially diluted and spotted on YPD plates. For hypoxic conditions, the plates were incubated at 30°C in BD GasPak EZ anaerobe pouches. The plates were observed and imaged for 2 d for aerobic conditions and 8 d for hypoxic conditions. The data were quantified as described (Petropavlovskiy et al, 2020). Briefly, the mean gray value quantitation was performed using ImageJ. The background for each image was adjusted using the sliding paraboloid function. The mean gray value was calculated for one dilution across multiple strains and background subtraction was performed.

**Telomere Southern blot**

Whole cell extract from wild-type and *set4Δ* strains was made by bead beating in phenol–chloroform–isoamyl alcohol. The extract was treated with RNase A and DNA was precipitated using ethanol. Genomic DNA was digested using the restriction enzyme XhoI, extracted with phenol–chloroform–isoamyl alcohol, and precipitated with ethanol. Digested DNA was subjected to electrophoresis on a 0.8% agarose gel in 0.5X TBE, the DNA was denatured in-gel and transferred onto a Hybond N+ nylon membrane (Amersham). The membrane was hybridized with a biotin-conjugated telomere probe (5'-biotin-CACACCCACACCCACACC-3') and was imaged using a Chemiluminescent Nucleic Acid Detection Module Kit (Thermo Fisher Scientific) and a Li-Cor C-DiGit Chemiluminescent Western Blot scanner.

**Zymolyase sensitivity assay**

WT (yEG001) and *set4Δ* (yEG322) cells were diluted and grown overnight to mid-log phase ($OD_{600}$ ~ 0.4–0.8) in aerobic or hypoxic

conditions. Cells were collected and resuspended in 1 ml sorbitol buffer (1.2 M sorbitol, 0.1 M KP04, pH 7.5) with 5 $\mu$l 2-mercaptoethanol and 5 $\mu$l of 10 mg/ml 100T zymolyase. Cells were incubated at room temperature, with occasional rocking, and the $OD_{600}$ was measured every 5 min in 1% SDS. Time $_{50\%\ OD}$ was determined as the time elapsed for the cultures to reach 50% of the starting $OD_{600}$, indicating 50% digestion by zymolyase and generation of spheroplasts.

### Trypan blue staining of cell walls

Detection of cell walls using Trypan blue was performed as described previously (Liesche et al, 2015). Briefly, 1 ml of log phase ($OD_{600}$ ~ 0.6–0.8) culture under aerobic and hypoxic conditions was centrifuged at 8,600$g$ for 2 min. The cells were washed once in PBS and resuspended in 1 ml PBS. Trypan blue was added at a final concentration of 10 $\mu$g/ml. 5 $\mu$l of cells were observed on a slide with coverslip using a Leica SP5 confocal microscope. Image processing was performed using ImageJ. Staining intensity and cell size was determined using the 3-D object counter plug-in for ImageJ. The plugin determines the intensity at the cell perimeter by identifying the object, setting a threshold for stained versus background pixels, and calculates the mean gray area of the stained pixels as a measurement of staining intensity.

### Chromatin immunoprecipitation

Chromatin immunoprecipitation (chIP) was performed as described (Meluh & Broach, 1999; Liu et al, 2005; Jezek et al, 2017a; Jezek et al, 2017b). Briefly, cultures were diluted and grown overnight to mid-log phase ($OD_{600}$ ~0.4–0.8). Cultures were then fixed with 1% formaldehyde for 20 min (histone chIPs) or 45 min (FLAG-Set4 and Sir3-HA chIPs). For the Rpd3-FLAG chIPs, a double crosslinking strategy was used as previously reported (Zeng et al, 2006) to improve recovery of Rpd3-FLAG with chromatin. In this case, cells were harvested and resuspended in PBS. EGS (ethylene glycol bis [succinimidyl succinate]) was added to a final concentration of 1.5 mM and cells were fixed for 30 min. Then 1% formaldehyde was added and cells were incubated for an additional 30 min. Quenching was performed with 0.5 M Tris–HCl pH 7.5 for 10 min. Cells were pelleted and washed with TBS before lysis.

Whole cell extracts were made by bead beating and the chromatin was digested with micrococcal nuclease enzyme. The amount of chromatin used was 40 $\mu$g per IP (histone chIPs) and 100–300 $\mu$g per IP (FLAG-Set4, Sir3-HA, and Rpd3-FLAG chIPs). The antibodies were either pre-bound to protein A/G magnetic beads (Pierce) overnight (histone, Sir3-HA, and Rpd3-FLAG chIPs) or pre-conjugated anti-FLAG M2 magnetic beads (Sigma-Aldrich) were used (FLAG-Set4 chIPs). The beads were added to the extracts and rotated overnight at 4°C. Protein-DNA complexes were eluted using 1% SDS and 0.1 M NaHCO$_3$, cross-links were reversed, and samples were treated with proteinase K and RNase A. DNA was extracted with phenol-chloroform-isoamyl alcohol and precipitated using ethanol. qPCR was performed as described above using 0.5 $\mu$l chIP DNA per reaction and gene-specific primers (Table S4). Three technical replicates were performed for each qPCR reaction and a minimum of three biological replicates were performed for each chIP experiment. Percent input was calculated relative to 10% of the input.

The following antibodies were used for chIP: rabbit anti-H4K5ac (Cat. no. ab51997; Abcam), rabbit anti-H4K12ac (Cat. no. ABE532; EMD Millipore), rabbit anti-H4K16ac (Cat. no. 07-329; EMD Millipore), rabbit anti-H3 (Cat. no. ab1791; Abcam), rabbit anti-H3K4me3 (Cat. no. 39159; Active Motif), rabbit anti-H3K9ac (Cat. no. 06-942; EMD Millipore), mouse anti-H3K36me3 (Cat. no. 61021; Active Motif), mouse anti-FLAG (Cat. no. F1804; Sigma-Aldrich), and mouse anti-HA (Cat. no. 05-904; EMD Millipore).

## Data Availability

The raw and processed data for RNA-sequencing experiments are available on the Gene Expression Omnibus database at accession number GSE173901.

## Supplementary Information

## Acknowledgements

The authors thank all members of the Green lab for helpful discussions, technical assistance, and feedback on the manuscript. We thank Dr. Paul Kaufman for yeast strains and Dr. Philip Farabaugh for sharing equipment. We are also grateful to Tagide deCarvalho of the Keith R. Porter Imaging Facility for assistance with microscopy. This work was support by the National Institutes of Health (R01GM124342 to EM Green) and the National Research Foundation of Korea (NRF-2019H1D3A2A02102167 to D Park).

### Author Contributions

Y Jethmalani: conceptualization, data curation, formal analysis, investigation, methodology, and writing—original draft.
K Tran: conceptualization, investigation, and methodology.
MY Negesse: investigation and methodology.
W Sun: investigation and methodology.
M Ramos: formal analysis, investigation, visualization, and methodology.
D Jaiswal: investigation and methodology.
M Jezek: investigation and methodology.
S Amos: investigation and methodology.
EJ Garcia: investigation and methodology.
D Park: data curation, formal analysis, supervision, and methodology.
EM Green: conceptualization, data curation, formal analysis, supervision, funding acquisition, methodology, project administration, and writing—original draft, review, and editing.

### Conflict of Interest Statement

The authors declare that they have no conflict of interest.

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
