## [Reviewer comments · Life Science Alliance]

Life Science Alliance

Set4 regulates stress response genes and coordinates histone deacetylases within yeast subtelomeres

Yogita Jethmalani, Khoa Tran, Maraki Negesse, Winny Sun, Mark Ramos, Deepika Jaiswal, Meagan Jezek, Shandon Amos, Eric Garcia, DoHwan Park, and Erin Green

DOI: <https://doi.org/10.26508/lsa.202101126>

Corresponding author(s): Erin Green, University of Maryland, Baltimore County

Review Timeline:

Submission Date:	2021-05-31
Editorial Decision:	2021-07-13
Revision Received:	2021-09-04
Editorial Decision:	2021-09-24
Revision Received:	2021-09-28
Accepted:	2021-09-29

Scientific Editor: Novella Guidi

Transaction Report:

July 13, 2021

Re: Life Science Alliance manuscript #LSA-2021-01126-T

Erin M Green
University of Maryland Baltimore County
Biological Sciences
Baltimore, MD 21250

Dear Dr. Green,

Thank you for submitting your manuscript entitled "Set4 regulates stress response genes and coordinates histone deacetylases within yeast subtelomeres" to Life Science Alliance. The manuscript was assessed by expert reviewers, whose comments are appended to this letter. As you will note from the reviewers' comments below, all the reviewers are quite interested and see the importance of these findings for the community. Besides Reviewer 2 which is quite positive, the others do raise important concerns regarding proper quantifications and analysis and as a main concern the lack of clear correlation between Set4-dependent gene expression, Set4 and deacetylase complexes binding to chromatin and histone acetylation/methylation marks. We, thus, encourage you to submit a revised version of the manuscript back to LSA that responds to all of the reviewers' points including evaluating the effect of SET4 overexpression to understand how it impacts growth and gene expression in hypoxic conditions, and assess whether the changes in expression levels observed on the specific genes correspond to complete mRNA expression or cryptic transcription by testing the H3K36me mark.

Thank you for this interesting contribution to Life Science Alliance. We are looking forward to receiving your revised manuscript.

Sincerely,

- A letter addressing the reviewers' comments point by point.
- An editable version of the final text (.DOC or .DOCX) is needed for copyediting (no PDFs).
- High-resolution figure, supplementary figure and video files uploaded as individual files: See our detailed guidelines for preparing your production-ready images, <https://www.life-science-alliance.org/authors>
- Summary blurb (enter in submission system): A short text summarizing in a single sentence the study (max. 200 characters including spaces). This text is used in conjunction with the titles of papers, hence should be informative and complementary to the title and running title. It should describe the context and significance of the findings for a general readership; it should be written in the present tense and refer to the work in the third person. Author names should not be mentioned.

B. MANUSCRIPT ORGANIZATION AND FORMATTING:

Reviewer #1 (Comments to the Authors (Required)):

In their work entitled "Set4 regulates stress response genes and coordinates histone deacetylases within yeast subtelomeres", Jethmalani et al. identified a role of SET4 in the regulation of subtelomeric genes expression comparing WT vs set4KO yeast strains by RNA-seq analysis. Following up using RTqPCR and chIP at specific telomeric regions, they noticed that Set4 function is strengthened under hypoxic stress. Finally, they showed a link between Set4KO phenotypes, histone acetylation level and Rpd3 and SIR complex recruitments to subtelomeric chromatin. This study is well described and uncover an important function for Set4, with potential interest in

other organisms with orthologs, such as the elusive SETD5 in human. However, there are certain points that remains unclear and need further investigation to finalize this work:

- Major comments

a) Based on specific genes expressions observed in Fig1&2 and further study in later figures, it seems that there are 2 classes of genes with different regulations: COS12 and YGL262W, which are positively regulated by Set4 in aerobic conditions, and YPS5, PAU11, PAU13, PAU21/22 which are negatively regulated by Set4, mostly under hypoxic condition. It is unclear why authors chose these genes, besides their subtelomeric localization. Were the first class part of the genes downregulated in the RNAseq analysis, and the second class ones upregulated by Set4 deletion? Were they within the top hits of deregulated genes? If not, it would be interesting to add few other genes to this study, representing either the up or down regulated class of genes from the RNA seq data.

b) Fig2A, why authors failed to show Set4KO aerobic vs hypoxic? This would indeed confirm if genes that are affected by Set4 are even more affected when it is highly expressed upon stress.

c) Fig4, how come H3K4me3 is not affected while gene expression is? If so, that would mean that changes in expression are not regulated by H3K4me3 but only by acetylation? This raises an important question: is the expression levels observed by authors on these specific genes corresponds to complete mRNA expression, or is it cryptic transcription? This could be easily answered by using other primers by RTqPCR or looking at the RNAseq data. Then, it has been shown that cryptic transcription is regulated by H3K36me. It would be interesting to test if this mark is affected or not.

d) In Fig 4 &5, SIR3 binding seems affected by Set4 deletion but not that much H4K16ac... In fact, one can see rather opposite effect between K16ac and SIR3 binding (for example comparing TEL07L TEL07L boundary). This is concerning and is another evidence that Set4 does not act on its own but rather participate in a more complex regulation mechanism. Also, why authors failed to show COS12 and YGL262W genes in this part of the study? It would be important in light of Fig 6, where it seems their regulation has no clear implication of Set4.

e) Fig 7, it is difficult to evaluate Set4 binding efficiency. As authors claim, there is a clear overexpression of SET4 under hypoxia and it is not possible to chip Set4 in basal condition. I find it hard to really see an enrichment compared to control CENVX. Can authors provide other evidence of Set4 specific binding to genes of interest compared to other control genes (unaltered in the RNAseq analysis for ex?) Also, is it possible to normalize to CENVX enrichment to have a better idea of binding specificity

f) Finally, the biggest concern of this study is the lack of clear correlation between Set4-dependent gene expressions (ERG3/11 expression not affected but clearly bound by Set4), Set4 and deacetylase complexes binding to chromatin (Sir2 or Rpd3 effects being independent of Set4 under some conditions) and histone acetylation/methylation marks. It seems that several classes of gene exist, with different regulation mechanisms involving Set4, with or without Sir3/Rpd3, with or without clear link to acetylation level, etc... This article is not defining such complex mechanisms, and it could be interesting to decipher a bit better Set4 mechanism: instead of total KO, can author repeat some of these experiments with specific SET4 PHD or SET domain mutants? At the very least, they should discuss this matter thoroughly. All this suggests that there is another important factor in the mix (one or several specific regulators) to modulate specifically some class of genes in aerobic

or hypoxic conditions. Based on Set4 existing interactomics in literature, can authors hypothesize some of these additional actors?

- Minor comments

a) Authors stated that "RNA-seq revealed an overall increase in differential gene expression between wildtype and set4^Δ cells in hypoxia, although we note that there was little change in the pattern of gene expression changes observed compared to aerobic conditions. Changes in gene expression of telomere-adjacent genes and the stress-induced PAU genes were observed under both normal and stress conditions." and "Set4 is lowly expressed under normal conditions, and its localization to these promoters was only detected during stress". It seems strange that SET4 deletion in basal condition is having such an effect in gene expression if it is poorly expressed. Even if there are more genes deregulated under hypoxia, authors should also provide information about the number of common genes regulated between normal and hypoxic conditions upon Set4 deletion (are they all shared or is there strong difference?), and among these common genes, provide information regarding a potential increase in fold change in hypoxia vs basal. Finally, a supplemental table with the top deregulated genes from the RNA seq analyses is needed.

b) It might be interesting to detail more in the text the SIR complex, as it is not clearly done in the actual manuscript and it can be difficult to understand the explanations switching from Sir2 to Sir3 related to Figure 5 results. (Sir2 being the HDAC and Sir3 being the recruiter of Sir2-Sir4 dimers to form the SIR complex).

c) I have a hard time to understand what author means in the sentence: "While we did not observe any differences in protein expression levels of Sir3-HA between wildtype and set4KO cells, nor between aerobic and hypoxic conditions (Figure S4B), the increased chromatin association of Sir3 in hypoxia in wildtype cells (Figure S5A), which is consistent with the global decrease in H4K16ac levels observed in hypoxia-treated cells (Figure S4A)."

Reviewer #2 (Comments to the Authors (Required)):

In this manuscript, the authors investigate the role of Set4 in regulating the expression of stress response genes within subtelomeric regions in budding yeast. They first found that the gene expression is disrupted in set4^Δ mutant cells, and among those up-regulated genes in set4^Δ mutant cells, there is an enrichment for genes involved in cell wall organization. They noted that some of the genes involved in cell wall organization are localized within subtelomeric regions and the expression of genes localized within subtelomeric regions is more likely to be changed than the ones in other regions in the absence of Set4. They then investigated two groups of subtelomeric genes: Group 1 is well-characterized for chromatin based silencing and telomere position effect (TPE), and Group 2 is PAU family genes, which are induced during anaerobic growth and are thought to be important for cell wall remodeling during stress. Intriguingly, the gene expression in group 1 is down-regulated in set4^Δ mutant cells, while the gene expression in group 2 is up-regulated in set4^Δ mutant cells.

Considering the previous reported role of Set4 under stress condition, they further subjected the cells to stress treatment. They found that upon hydrogen peroxide treatment, in the absence of Set4, the group 1 gene did not show substantial change, but group 2 gene are more highly up-regulated than under normal conditions. When subjected to hypoxia condition, they found again

that the group 2 genes were more highly upregulated in set4 mutant cells. Along with these gene expression changes, they observed a mild sensitivity to zymolyase digestion in hypoxia for set4 mutant cells, indicating a role of Set4 in cell -wall organization during hypoxia.

To investigate the mechanism underlying these observations, they analyzed the levels of different histone modifications and the chromatin binding of Set4, Rpd3 HDAC and Sir3 under stress treatment. They found that Set4 is recruited to the subtelomeric regions during hypoxia. In set4 mutant cells around PAU promoter regions, the levels of several histone acetylation marks are increased, and Sir3 binding is decreased. Moreover, Rpd3 HADC is also decreased in set4 mutant cells around PAU promoter regions. Together, they propose a model for Set4 function in subtelomeric gene expression regulation during hypoxic stress.

In general, the results provide new information regarding the role of Set4 during stress response. To increase the strength of this work, it would better to investigate more the connections between Set4 recruitment and the SIR3 and/or Rpd3 chromatin binding. I'm wondering how Set4 is recruited to the subtelomeric regions? How does Set4 promote the SIR proteins' and/or Rpd3's chromatin binding?

Reviewer #3 (Comments to the Authors (Required)):

Review of LSA-2021-01126-T 'Set4 regulates stress response genes and coordinates histone deacetylases within yeast subtelomeres'

This is an interesting study where the authors follow-up a connection of the chromatin-associated protein Set4 on the expression of genes important in cell wall homeostasis. It is overall well-written and consistent but the phenotypic/physiological data are unfortunately not convincing in their present state. This needs to be addressed for the manuscript to merit being published in LSA. In addition, given the multiple parallels the authors make to their previous study investigating Set4 effects under hydrogen peroxide stress, it would be informative if the authors would assay the effect of SET4 overexpression, making cells resistant to hydrogen peroxide, on hypoxic growth and subtelomeric gene expression.

Major comments:

P11, lines 2-4: given the beneficial effect of SET4 overexpression on hydrogen peroxide resistance it would be interesting to know how SET4 overexpression impacts growth in hypoxic conditions and eg gene expression.

P11, lines 5-7: From the growth data in Fig. 3A it seems the set4 strain indeed grows slower upon hypoxia and that this defect can be rescued by 1.5 M sorbitol, but the data needs to be repeated, quantified properly and subjected to statistical analyses.

P11, lines 18-26: the description of the data obtained by TB cell wall staining does not reflect the data presented in Fig. 3C. From Fig 3C it seems like TB staining decreases to a larger extent in the set4 mutant than in the wt, whereas the text at p11 indicates the opposite. In addition, it is not clear why the cell wall staining increases in set4 mutant cells growing aerobically?

More specifically, it is unclear how the decreased 'Mean grey area' signal the authors measured, takes into account the general decrease in cell size observed by confocal microscopy upon hypoxia

in the methods paper cited (Liesche J et al, Front Microbiol, 2015)? In this paper the most discriminative measurement for cell wall deficiency seems to be the ratio total cell wall volume vs total cell volume and a measurement specifically of this parameter might clarify the discrepancy between the results of the cell wall staining presented with those of eg the gene expression and the zymolyase digestion sensitivity. This apparent paradox would be important to provide further support for Set4 affecting cell wall morphology rather than generally affecting e.g. cell size and other parameters.

P13, lines 12-20: the growth data on interactions between Sir2 / Rpd3 and Set4 under hypoxia shown in Fig. S5 are very interesting but would be more convincing if they could be measured quantitatively and presented with proper statistical analysis. Quantitative data on phenotypic interactions between Sir2/Rpd3 and Set4 would fit among the figures proper not just in the supplement.

Minor comments:

Figure 3C, the cells in the TB staining are very hard to see. It would be better if the images were zoomed-in on a handful of representative cells instead.

Unusual characters, mainly delta signs, are misformatted in most of the figures (Figs 3,4,5 and 6), could they be arranged to standard fonts?

We appreciate the feedback and comments provided by the reviewers and editor. We are pleased that all three reviewers found the paper of interest and that it made new contributions to the field. Our response to the reviewer comments is detailed point-by-point below.

Response to reviewer #1:

In their work entitled "Set4 regulates stress response genes and coordinates histone deacetylases within yeast subtelomeres", Jethmalani et al. identified a role of SET4 in the regulation of subtelomeric genes expression comparing WT vs set4KO yeast strains by RNA-seq analysis. Following up using RTqPCR and ChIP at specific telomeric regions, they noticed that Set4 function is strengthened under hypoxic stress. Finally, they showed a link between Set4KO phenotypes, histone acetylation level and Rpd3 and SIR complex recruitments to subtelomeric chromatin. This study is well described and uncover an important function for Set4, with potential interest in other organisms with orthologs, such as the elusive SETD5 in human.

We appreciate the interest and positive feedback of the reviewer. We have addressed specific concerns of the reviewer below.

1-a) Based on specific genes expressions observed in Fig1&2 and further study in later figures, it seems that there are 2 classes of genes with different regulations: COS12 and YGL262W, which are positively regulated by Set4 in aerobic conditions, and YPS5, PAU11, PAU13, PAU21/22 which are negatively regulated by Set4, mostly under hypoxic condition. It is unclear why authors chose these genes, besides their subtelomeric localization. Were the first class part of the genes downregulated in the RNAseq analysis, and the second class ones upregulated by Set4 deletion? Were they within the top hits of deregulated genes? If not, it would be interesting to add few other genes to this study, representing either the up or down regulated class of genes from the RNA seq data.

The genes chosen for further analysis by RT-qPCR were top hits in the RNA-seq analysis. As the reviewer states, these genes were chosen because they represented the two major classes of subtelomeric genes identified in the RNA-seq: upregulated and downregulated in the absence of Set4. The top 10% of genes under aerobic and hypoxic conditions are now highlighted in Table S1 (see last two tabs in the Excel file). We observed that the PAU gene family in particular was highly enriched in the differentially expressed gene sets under both aerobic and hypoxic conditions, therefore we analyzed expression of multiple PAU genes residing on different chromosomes. In addition, as we describe in the text, we observed enrichment for altered expression at other subtelomeric genes, including COS12, YGL262W, and YPS5. These were chosen for further analysis as they span the subtelomeric region of Chr07L, which also contains PAU11. The genes chosen for analysis here represent the major categories of differentially-expressed genes that we identified, which include genes linked to cell wall

organization (see Table 1) and subtelomerically-encoded genes (Figures 1A, 1B, 2E, 2F). We have further clarified this rationale in the text on the bottom of page 4.

1-b) Fig2A, why authors failed to show Set4KO aerobic vs hypoxic? This would indeed confirm if genes that are affected by Set4 are even more affected when it is highly expressed upon stress.

Due to the magnitude of the differences in expression of some of the target genes tested in aerobic versus hypoxic conditions, we are not able to easily depict these data on the same graph. This is evident in Figure 2A, in which we demonstrate that there is more than 200-fold increase in expression in the *PAU* genes in hypoxic conditions compared to aerobic conditions. Therefore, Figure 1C shows the difference between wildtype and *set4Δ* cells under aerobic conditions and Figure 2B shows this difference in hypoxia. We did not combine these data on to one graph because of the very large difference in expression in these genes under aerobic and hypoxic conditions even in wildtype cells. Even with an axis break, the differences in expression levels between wildtype and mutant are difficult to visualize with this range in the data.

1-c) Fig4, how come H3K4me3 is not affected while gene expression is? If so, that would mean that changes in expression are not regulated by H3K4me3 but only by acetylation? This raises an important question: is the expression levels observed by authors on these specific genes corresponds to complete mRNA expression, or is it cryptic transcription? This could be easily answered by using other primers by RTqPCR or looking at the RNAseq data. Then, it has been shown that cryptic transcription is regulated by H3K36me. It would be interesting to test if this mark is affected or not.

To address this concern, we have added new data to the manuscript investigating a role for H3K36me3-mediated regulation of cryptic transcription at the *PAU* gene loci and within the subtelomere. These new data, as well as some previous observations, suggest that there is not a major role for Set2-repressed cryptic transcripts at the gene loci tested. Our observations supporting this conclusion are summarized below:

1. We performed chIP of H3K36me3 across the subtelomere and at *PAU* promoters and ORFs. We did not observe any substantial differences between the levels of this mark in wildtype and *set4Δ* cells under aerobic or hypoxic conditions. These data are shown in Figure 5B.

2. We monitored expression of the *PAU* genes and other subtelomeric genes in *set2Δ* cells using the same sets of RT-qPCR primers as used for the *set4Δ* mutants. We observed a slight decrease in expression levels of the *PAU* genes in *set2Δ* cells under both aerobic and hypoxic conditions. However, if Set2-repressed cryptic transcripts were being produced at these loci and measured by our primers, it is expected that that RT-qPCR signal would be increased in the absence of Set2. This indicates that the Set2-repressed cryptic transcripts are not a contributor to the mRNA products that we are monitoring in our RT-qPCR. These data are shown in Supplemental Figure 4.

3. H3K36 methylation by Set2 is known to recruit Rpd3 to promote cryptic transcript repression (Keogh et al., *Cell*, 2005; Carozza et al., *Cell*, 2005 and others). Cryptic transcripts are therefore expected to increase in the absence of Rpd3. Our RT-

qPCR data of *rpd3Δ* cells shows decreased expression of all of the subtelomeric genes tested in these mutants. This further suggests that our RT-qPCR primers are not assaying cryptic transcripts in the regions, but correspond to the expression of *PAU* gene mRNAs. The *rpd3Δ* gene expression data are in Figure 7C.

4. We do not see evidence of cryptic transcripts being produced in the absence of *Set4* in our RNA-seq data at the subtelomeric regions tested here. One complication of this analysis is that there is highly repetitive sequence in the regions, and only uniquely-mapped reads were analyzed in the RNA-seq. Therefore, this analysis may not fully represent the extent of transcription in the area and we have not focused on it in the manuscript for this reason. However, combined with the other genetic tools that we have employed, our data indicate that the gene expression changes described here are not due to measurement of cryptic transcripts in *set4Δ* cells but represent coding mRNAs transcribed from the subtelomeric genes.

1-d) In Fig 4 & 5, SIR3 binding seems affected by Set4 deletion but not that much H4K16ac... In fact, one can see rather opposite effect between K16ac and SIR3 binding (for example comparing TEL07L TEL07L boundary). This is concerning and is another evidence that Set4 does not act on its own but rather participate in a more complex regulation mechanism. Also, why authors failed to show COS12 and YGL262W genes in this part of the study? It would be important in light of Fig 6, where it seems their regulation has no clear implication of Set4.

Based on the histone deacetylase activity and known roles for chromatin regulation of the SIR complex at telomeres, our chIP data of H4K16ac and Sir3 protein abundance are consistent with each another. At *TEL07L*, SIR complex binding is enriched near the chromosome end and this increased binding leads to lower levels of H4K16ac (they are inversely-related). In the absence of the *Set4*, we observe decreased SIR complex binding (Figure 6A). We also see a slight increase in H4K16ac levels in *set4Δ* cells in hypoxia (Figure 4) at the *TEL07L* boundary, consistent with decreased SIR binding in the region.

As described on the top of page 5, *COS12* and *YGL262W* are located within the *TEL07L* subtelomeric region. The chIP primers used are adjacent to these genes, with one set of primers representing the highly-repressed telomere end (*TEL07L* primer set) and one set of primers annealing near the boundary between this silent, repressed region with more highly-expressed euchromatin. This primer location is described on the top of page 8. These primer locations are frequently used to assess the distribution of histone modifications and modifying-enzymes in distinct regions of the subtelomere.

1-e) Fig 7, it is difficult to evaluate Set4 binding efficiency. As authors claim, there is a clear overexpression of SET4 under hypoxia and it is not possible to chip Set4 in basal condition. I find it hard to really see an enrichment compared to control CENVX. Can authors provide other evidence of Set4 specific binding to genes of interest compared to other control genes (unaltered in the RNAseq analysis for ex?) Also, is it possible to normalize to CENVX enrichment to have a better idea of binding specificity

We have now shown the signal for Set4-FLAG chIP at another negative control region. We used primers that anneal to the coding sequence of the gene *PRP8*, which shows no change in expression in the RNA-seq. These primers also show very little signal for Set4-FLAG compared to the untagged control. Furthermore, we think that showing the percent input for the chIP is the most accurate representation of the data, as it more directly demonstrates localization at different chromosomal regions rather than normalizing these data to a negative control region.

1-f) Finally, the biggest concern of this study is the lack of clear correlation between Set4-dependent gene expressions (ERG3/11 expression not affected but clearly bound by Set4), Set4 and deacetylase complexes binding to chromatin (Sir2 or Rpd3 effects being independent of Set4 under some conditions) and histone acetylation/methylation marks. It seems that several classes of gene exist, with different regulation mechanisms involving Set4, with or without Sir3/Rpd3, with or without clear link to acetylation level, etc... This article is not defining such complex mechanisms, and it could be interesting to decipher a bit better Set4 mechanism: instead of total KO, can author repeat some of these experiments with specific SET4 PHD or SET domain mutants? At the very least, they should discuss this matter thoroughly. All this suggests that there is another important factor in the mix (one or several specific regulators) to modulate specifically some class of genes in aerobic or hypoxic conditions. Based on Set4 existing interactomics in literature, can authors hypothesize some of these additional actors?

We agree with the reviewer that there is likely a complex set of interactions that are controlling expression of subtelomeric genes under normal and stress conditions in Set4-mediated pathways. The regulation of subtelomeric chromatin relies on a large set of chromatin proteins, including histone deacetylases, acetyltransferases, methyltransferases, and transcription factors (particularly at the local stress response genes). Disruption of one of these factors within the subtelomeric region or adjacent euchromatin can have both direct and indirect consequences on gene regulation. Our model indicates that the changes in gene expression observed in the absence of Set4 are likely due to both direct effects of Set4 and indirect effects on the regional chromatin environment. To better specify some of the gene regulatory mechanisms involved, we have tested the dependence of Set4 regulation of subtelomeric genes on the Upc2 transcription factor, a known regulator of anaerobic response genes. These data are provided in Figure 8 and we have discussed the interaction between Upc2 and Set4 on the top of page 11 (Results) and the bottom of page 13 (Discussion).

While we agree with the reviewer that better defining the role of the both the PHD finger and the SET domain in Set4-dependent gene regulation is a critical question, the experiments as described are currently not feasible or beyond the scope of this work. We have found that deletion of the SET domain in Set4 substantially destabilizes the protein, such that any studies of this allele of *SET4* are not likely to just represent the role of the SET domain. Furthermore, given the limited understanding of the divergent SET domain of Set4, point mutations and deletions within the domain are of little use as it is now known whether any particular functions are being disrupted. The PHD finger of Set4 has been studied in other contexts and it does not appear to bind histones directly

(Shi *et al.*, *JBC*, 2007; Gatchalian *et al.*, *JMB*, 2017), as is well-characterized for other PHD fingers. Therefore, the function of the PHD finger in Set4 is not known, and deleting this domain without understanding its role in the protein will also not be useful to determining the mechanism until much more is known about the PHD finger. This line of experiments is beyond the scope of the current work.

- *Minor comments*

1-a) Authors stated that "RNA-seq revealed an overall increase in differential gene expression between wildtype and set4Δ cells in hypoxia, although we note that there was little change in the pattern of gene expression changes observed compared to aerobic conditions. Changes in gene expression of telomere-adjacent genes and the stress-induced PAU genes were observed under both normal and stress conditions." and "Set4 is lowly expressed under normal conditions, and its localization to these promoters was only detected during stress". It seems strange that SET4 deletion in basal condition is having such an effect in gene expression if it is poorly expressed. Even if there are more genes deregulated under hypoxia, authors should also provide information about the number of common genes regulated between normal and hypoxic conditions upon Set4 deletion (are they all shared or is there strong difference?), and among these common genes, provide information regarding a potential increase in fold change in hypoxia vs basal. Finally, a supplemental table with the top deregulated genes from the RNA seq analyses is needed.

As requested by the reviewer, we have provided tables with the top 10% of differentially-expressed genes for aerobic and hypoxic conditions as part of Table S1 (please see the last two tabs). In terms of comparing the differentially-expressed genes under both conditions, our primary point is that the same categories of genes are misregulated in aerobic and hypoxic conditions. This is highlighted in Table 1, in which genes related to cell wall organization are misregulated under both aerobic and hypoxic conditions upon loss of Set4. This is also indicated by our analysis of gene location, which showed enrichment for differential expression of subtelomeric genes as shown in Figures 1A and 1B (aerobic) and 2E and 2F (hypoxic). Furthermore, we have chosen two representative classes of these genes to test extensively using RT-qPCR. We have now emphasized in the text on the bottom of page 4 that this conclusion is based on categories of genes, rather than a gene-by-gene comparison. Furthermore, a gene-by-gene comparison of the RNA-seq data may be a bit misleading, as some of the genes are so lowly expressed under aerobic conditions that they are automatically excluded from the analysis (due to very low gene hit counts). However, visual inspection of the genes shown in the tables of top 10% of differentially expressed genes clearly demonstrates that *PAU* genes are misregulated under both aerobic and hypoxic conditions, along with other subtelomeric loci.

We agree with the reviewer that it is somewhat unusual that Set4 has an effect on gene expression in basal conditions given its low expression, however there are multiple examples of lowly-expressed transcriptional regulators, and other published work has shown similar changes in expression in *set4Δ* cells under non-stressed conditions

(Kemmeren *et al.*, *Cell*, 2014). Furthermore, our RNA-seq and RT-qPCR data clearly emphasize the differential expression changes are overall greater in hypoxia compared to aerobic conditions (see Table 1, for example).

1-b) *It might be interesting to detail more in the text the SIR complex, as it is not clearly done in the actual manuscript and it can be difficult to understand the explanations switching from Sir2 to Sir3 related to Figure 5 results. (Sir2 being the HDAC and Sir3 being the recruiter of Sir2-Sir4 dimers to form the SIR complex).*

We have added this information to the text on page 9 in the second paragraph of the Results section describing the ChIP experiments.

1-c) *I have a hard time to understand what author means in the sentence: "While we did not observe any differences in protein expression levels of Sir3-HA between wildtype and set4KO cells, nor between aerobic and hypoxic conditions (Figure S4B), the increased chromatin association of Sir3 in hypoxia in wildtype cells (Figure S5A), which is consistent with the global decrease in H4K16ac levels observed in hypoxia-treated cells (Figure S4A)."*

We apologize for the confusion in this sentence and have rewritten this part.

Response to Reviewer 2:

We thank the reviewer for their comments and appreciate their overall interest in the work. We have addressed the main set of questions asked by the reviewer below.

2-a) *In general, the results provide new information regarding the role of Set4 during stress response. To increase the strength of this work, it would better to investigate more the connections between Set4 recruitment and the SIR3 and/or Rpd3 chromatin binding. I'm wondering how Set4 is recruited to the subtelomeric regions? How does Set4 promote the SIR proteins' and/or Rpd3's chromatin binding?*

We agree with the reviewer that these are interesting questions that would provide more mechanistic insight into the role of Set4 at subtelomeres. In our revised manuscript, we have provided some new data on the genetic interaction between Set4 and the transcription factor Upc2 in regulating gene expression. Our current model (Figure 10 and as described in the Discussion) suggests a regional chromatin regulatory role for Set4 within subtelomeres, which appears most critical to maintain proper levels of *PAU* gene expression. It is unlikely that Set4 directly recruits the SIR proteins or Rpd3, and their mechanisms of chromatin localization are different, so it is not expected that Set4 would directly regulate recruitment of these factors. In terms of Set4 recruitment to chromatin, this could be mediated by the PHD finger or protein interactors of Set4. As described in response to Reviewer 1, point 1-f, there is little understanding of the Set4 PHD finger and further investigation is outside the scope of this work. Overall, we agree with the reviewer that these are important and intriguing questions to address, however,

due to the complexity of interactions among chromatin regulators at the subtelomere and our still limited knowledge of Set4 biochemical functions, this would greatly increase the scope of this work. Nonetheless, our study provides important new insights into the biological role for Set4 under normal and stress conditions, which is a critical foundation for future investigation of its detailed molecular roles at chromatin.

Response to Reviewer #3:

This is an interesting study where the authors follow-up a connection of the chromatin-associated protein Set4 on the expression of genes important in cell wall homeostasis. It is overall well-written and consistent but the phenotypic/physiological data are unfortunately not convincing in their present state. This needs to be addressed for the manuscript to merit being published in LSA. In addition, given the multiple parallels the authors make to their previous study investigating Set4 effects under hydrogen peroxide stress, it would be informative if the authors would assay the effect of SET4 overexpression, making cells resistant to hydrogen peroxide, on hypoxic growth and subtelomeric gene expression.

We thank the reviewer for their positive comments and interest in our work. We have addressed the reviewer's concerns in our revised manuscript and provide a response to those concerns below.

Major comments:

3-a) P11, lines 2-4: given the beneficial effect of SET4 overexpression on hydrogen peroxide resistance it would be interesting to know how SET4 overexpression impacts growth in hypoxic conditions and eg gene expression.

This is an interesting question raised by the reviewer. In our previous work (Tran *et al.*, *JBC*, 2018), we demonstrated that under unstressed, aerobic conditions, overexpression of *SET4* caused decreased cell growth. We used an overexpression system in which *SET4* was induced from a modified *GAL1* promoter that responds to an artificial transcription factor induced by beta-estradiol (Mclsaac *et al.*, *NAR*, 2014). We previously showed that low level overexpression of Set4 prior to treatment with hydrogen peroxide improved survival following the stress treatment. In the original version of this manuscript, we did not include data regarding overexpression of Set4 in hypoxia because there are a number of differences in the biology and the experimental setup which make it hard to draw parallels between the two stress conditions. First, for oxidative stress, we used an acute stress of 30 min treatment with hydrogen peroxide. For hypoxia, the timeframe we used most commonly in this manuscript is 18 hours growth in hypoxic conditions in liquid culture or up to 8 days on plates. Furthermore, we do not think there is much change in Set4 protein levels in 30 min of oxidative stress treatment, whereas in hypoxia, Set4 levels increase dramatically. This increase in abundance is already more substantially more Set4 than in aerobic cells, so it is not clear that adding even more Set4 to hypoxic cells would have much impact. Finally, extended overexpression of Set4 inhibits cell growth, which can confound the interpretation of the experiment. We were able to overcome this problem in oxidative

stress by using a very small amount of inducer and short time frames. However, with the long timeframes for incubation in hypoxia, it is difficult to ensure that Set4 is continuously being overexpressed, particularly at low levels of inducer. As requested by the reviewer and described below, we have tested whether Set4 overexpression impacts growth in hypoxia (Reviewer Figure 1A) and whether overexpression leads to changes in *PAU* gene expression under aerobic conditions (Reviewer Figure 1B). In the plate growth assay (1A), we do not observe a clear advantage for increased levels of Set4 in hypoxic cells in promoting cell growth. At two different concentrations of beta-estradiol, we observe inhibition of cell growth with *SET4* overexpression (compared to empty vector- EV). This phenotype is similar in hypoxic cells. We also did not observe changes to *PAU* gene expression levels when Set4 is overexpressed under aerobic conditions (1B).

We have not included these data in the manuscript because we did not observe much of an impact of Set4 overexpression on these phenotypes. Furthermore, the multiple differences between the oxidative stress and hypoxic conditions suggest to us that the impacts of Set4 overexpression are not likely to parallel one another under the two conditions due to both biological and technical factors.

[Figure removed by editorial staff per authors' request].

3-b)P11, lines 5-7: From the growth data in Fig. 3A it seems the set4 strain indeed grows slower upon hypoxia and that this defect can be rescued by 1.5 M sorbitol, but the data needs to be repeated, quantified properly and subjected to statistical analyses.

The assay we presented is quantitative in that cells are plated and serially diluted in set amounts so that growth and colony size can be compared between strains. In this figure, 10-fold serial dilutions were used, which allow us to estimate that *set4Δ* mutants grow at least 10-fold slower than wildtype cells, in addition to the apparent smaller colony size evidenced on the plate. We have repeated this assay with several isolates of *set4Δ* mutants and have consistently observed a decrease in growth in *set4Δ* cells in hypoxia compared to wildtype. (This can also be observed with additional isolates in Figure 7A). We used a previously published image analysis method to more quantitatively show this growth difference (Petropavlovskiy *et al.*, *STAR Protoc.*, 2020), which can be seen in Figure S5B. In addition, we repeated the growth assay on

sorbitol-containing medium as well. While we did observe mildly improved growth of *set4Δ* mutants in sorbitol, the effect size was small and our quantitation was not statistically significant. We have therefore removed this data from the manuscript.

3-c) P11, lines 18-26: the description of the data obtained by TB cell wall staining does not reflect the data presented in Fig. 3C. From Fig 3C it seems like TB staining decreases to a larger extent in the set4 mutant than in the wt, whereas the text at p11 indicates the opposite. In addition, it is not clear why the cell wall staining increases in set4 mutant cells growing aerobically?

We are not clear where the conclusion that the TB staining decreases in the *set4Δ* mutant compared to wildtype was drawn from. In the images provided, it is evident that the intensity of the signal decreases in wildtype cells grown in hypoxic compared to aerobic conditions. In the *set4Δ* mutant, the cells grown in hypoxia show an increased intensity of TB signal compared to hypoxic wildtype cells and the intensity appears more similar to aerobically-grown cells. As described below, we have now presented a new analysis method in which the mean intensity of the signal at the cell perimeter was determined (which is a cell-size independent parameter) and this also shows that there is more TB signal in *set4Δ* mutants than wildtype cells than hypoxia. In addition, with this method, we did not observe a statistically-significant difference between wildtype and *set4Δ* mutants in aerobic conditions.

3-d) More specifically, it is unclear how the decreased 'Mean grey area' signal the authors measured, takes into account the general decrease in cell size observed by confocal microscopy upon hypoxia in the methods paper cited (Liesche J et al, Front Microbiol, 2015)? In this paper the most discriminative measurement for cell wall deficiency seems to be the ratio total cell wall volume vs total cell volume and a measurement specifically of this parameter might clarify the discrepancy between the results of the cell wall staining presented with those of eg the gene expression and the zymolyase digestion sensitivity. This apparent paradox would be important to provide further support for Set4 affecting cell wall morphology rather than generally affecting e.g. cell size and other parameters.

To address this concern of the reviewer, we have used an alternate analysis method in ImageJ with the 3D object counter plugin. This provided a measurement of the mean intensity of the trypan blue staining around the cell perimeter. This measurement is independent of cell size and is therefore more reflective of the cell wall composition. Analyses of these data revealed that TB staining is less intense in both wildtype and *set4Δ* mutants in hypoxia compared to aerobic conditions (see Figure 3D). However, *set4Δ* cells show increased TB intensity compared to wildtype cells in hypoxia, indicating altered cell wall remodeling in *set4Δ* mutants in hypoxia. Our analyses also measured cell surface area, which did not reveal a significant change in cell size in hypoxic wildtype cells compared to aerobic wildtype, though this may be because we did not measure cell volume, as reported in the cited reference. In addition, we did not observe any statistically-significant differences in cell size between wildtype and *set4Δ* mutants.

Although the reviewer states that there is a paradox regarding the reduced zymolyase sensitivity of *set4Δ* mutants in hypoxia and the increased TB stain in hypoxia, these assays monitor different aspects of the cell wall and the pattern that we observe is in line with previously published observations. The TB stain is an indication of the chitin and beta-glucan mass in the cell wall, whereas the zymolyase digestion assay is most sensitive to the extent of crosslinking, most likely of beta 1,6-glucans. Acquilar-Uscanga and Francois (*Lett. Appl. Microbiol.*, 2003) showed that there are specific changes in cell wall composition in low oxygen conditions. This includes an overall reduction of cell wall mass by 25%, including a three-fold decrease in chitin although little change in mannans and beta-glucan composition. While they did not test zymolyase sensitivity under their culture conditions, the increased resistance to zymolyase in hypoxia has been demonstrated by Liesche et al, 2015 and others. The phenotypes we observe in *set4Δ* mutants are therefore consistent with an attenuation of cell wall remodeling in response to hypoxia in *set4Δ* mutants both in the zymolyase sensitivity assay and the TB staining assay, suggesting that these assays are consistent with one another. There is still very little understanding of the precise molecular role of the *PAU* gene products in cell wall structure and organization, therefore it is difficult to draw direct connections between *PAU* gene expression levels and cell wall organization at this stage.

3-e) P13, lines 12-20: the growth data on interactions between Sir2 / Rpd3 and Set4 under hypoxia shown in Fig. S5 are very interesting but would be more convincing if they could be measured quantitatively and presented with proper statistical analysis. Quantitative data on phenotypic interactions between Sir2/Rpd3 and Set4 would fit among the figures proper not just in the supplement.

We have added these data to the main figures, now shown in Figure 7A. While quantitative conclusions can be drawn from the growth on plates, we have performed quantitation as described above and in Petropavlovskiy *et al.*, *STAR Protoc.*, 2020. These data are shown in Figure S5B.

Minor comments:

3-f) Figure 3C, the cells in the TB staining are very hard to see. It would be better if the images were zoomed-in on a handful of representative cells instead.

We have now provided zoomed-in images of just part of the field in Figure 3C.

3-g) Unusual characters, mainly delta signs, are misformatted in most of the figures (Figs 3,4,5 and 6), could they be arranged to standard fonts?

We apologize for this and have corrected it in the new figures.

September 24, 2021

RE: Life Science Alliance Manuscript #LSA-2021-01126-TR

Dr. Erin M Green
University of Maryland, Baltimore County
Biological Sciences
1000 Hilltop Circle
Baltimore, MD 21250

Dear Dr. Green,

Thank you for submitting your revised manuscript entitled "Set4 regulates stress response genes and coordinates histone deacetylases within yeast subtelomeres". We would be happy to publish your paper in Life Science Alliance pending final revisions necessary to meet our formatting guidelines.

- please upload your supplementary figures as single files also
- please incorporate the Supplemental References into the main Reference list
- please consult our manuscript preparation guidelines <https://www.life-science-alliance.org/manuscript-prep> and make sure your manuscript sections are in the correct order
- please add an Author Contributions section to your main manuscript text
- please add a conflict of interest statement to your main manuscript text
- please add your main, supplementary figure, and table legends to the main manuscript text after the references section
- there is a call-out for figure S4A, although the legend and the actual figure have no labeled panels. Please revise. The same for a call-out for figure 5D.

A. FINAL FILES:

B. MANUSCRIPT ORGANIZATION AND FORMATTING:

Sincerely,

Reviewer #1 (Comments to the Authors (Required)):

I would like to thank authors for their precise responses and their effort in order to improve the quality of their study. I am satisfied with revisions made by authors in the updated version of this manuscript and I do not request any additional experiment or discussion and consider the manuscript publishable as it is.

Reviewer #2 (Comments to the Authors (Required)):

The revised manuscript has been greatly improved. I am satisfied with the authors' response to my comments.

Reviewer #3 (Comments to the Authors (Required)):

In the revised version the authors have overall done a good job in amending the comments I had. I think it is now ready for publication in LSA.

September 29, 2021

RE: Life Science Alliance Manuscript #LSA-2021-01126-TRR

Dr. Erin M Green
University of Maryland, Baltimore County
Biological Sciences
1000 Hilltop Circle
Baltimore, MD 21250

Dear Dr. Green,

Thank you for submitting your Research Article entitled "Set4 regulates stress response genes and coordinates histone deacetylases within yeast subtelomeres". It is a pleasure to let you know that your manuscript is now accepted for publication in Life Science Alliance. Congratulations on this interesting work.

DISTRIBUTION OF MATERIALS:

Again, congratulations on a very nice paper. I hope you found the review process to be constructive and are pleased with how the manuscript was handled editorially. We look forward to future exciting submissions from your lab.

Sincerely,
